# Enhancing Statistical Validity and Power in Hybrid Controlled Trials: A Randomization Inference Approach with Conformal Selective Borrowing

**Ke Zhu** [1 2]  **Shu Yang** [1]  **Xiaofei Wang** [2]

## Abstract

External controls from historical trials or observational data can augment randomized controlled trials when large-scale randomization is impractical or unethical, such as in drug evaluation for rare diseases. However, non-randomized external controls can introduce biases, and existing Bayesian and frequentist methods may inflate the type I error rate, particularly in small-sample trials where external data borrowing is most critical. To address these challenges, we propose a randomization inference framework that ensures finite-sample exact and model-free type I error rate control, adhering to the "analyze as you randomize" principle to safeguard against hidden biases. Recognizing that biased external controls reduce the power of randomization tests, we leverage conformal inference to develop an individualized test-then-pool procedure that selectively borrows comparable external controls to improve power. Our approach incorporates selection uncertainty into randomization tests, providing valid post-selection inference. Additionally, we propose an adaptive procedure to optimize the selection threshold by minimizing the mean squared error across a class of estimators encompassing both no-borrowing and full-borrowing approaches. The proposed methods are supported by non-asymptotic theoretical analysis, validated through simulations, and applied to a randomized lung cancer trial that integrates external controls from the National Cancer Database.

## 1. Introduction

Randomized controlled trials (RCTs) are the gold standard for making causal inferences on the treatment effect of a new treatment relative to a control treatment. However, large RCTs are often infeasible to conduct in practice when the indications of interest involve rare diseases (U.S. Food and Drug Administration, 2022) or common conditions where few patients are willing to participate due to a lack of equipoise (Miller & Joffe, 2011). RCTs in such a context often lack sufficient statistical power to detect realistic treatment effect sizes. Meanwhile, historical studies or large external databases provide real-world data under control conditions, often referred to as external controls (ECs). By integrating RCT with ECs, hybrid controlled trials have garnered significant interest as an effective approach to enhance the power of RCTs with small sample sizes. However, most existing methods for hybrid controlled trials rely on model-based or asymptotic $p$-values, which can lead to inflated type I error rates when the randomized sample size is small, or the model is misspecified. Moreover, since ECs are not randomized, they may systematically differ from randomized controls, even after adjusting for measured confounders. Directly incorporating these ECs may introduce hidden bias, compromising the validity of the statistical inference. Strictly controlling the type I error rate in hybrid controlled trials, especially with small sample sizes and unmeasured confounding, remains an open problem.

To address this problem, we extend the randomization inference framework to hybrid controlled trials. To utilize ECs, we use a doubly robust estimator of the average treatment effect (ATE) as the test statistic, which incorporates both RCT and EC data and effectively balances the measured confounders between RCT and EC (Li et al., 2023b). Then, Fisher randomization tests (FRTs) are performed using only the randomization in the RCT. In contrast to the asymptotic inference in Li et al. (2023b), which relies on (i) large sample sizes for both the RCT and EC, (ii) correct specification of at least one of the two nuisance models, and (iii) no unmeasured confounders, the FRT strictly controls the type I error rate without requiring any of these conditions, thus achieving model-free, finite-sample exact inference. The validity of the FRT relies solely on the randomization

---

[1]Department of Statistics, North Carolina State University, Raleigh, NC 27695, U.S.A. [2]Department of Biostatistics and Bioinformatics, Duke University, Durham, NC 27710, U.S.A.. Correspondence to: Shu Yang <syang24@ncsu.edu>.

*Proceedings of the 42$^{nd}$ International Conference on Machine Learning*, Vancouver, Canada. PMLR 267, 2025. Copyright 2025 by the author(s).

within the RCT, which is typically well-managed by the study design. Furthermore, we perform a power analysis for FRT in hybrid controlled trials and show that incorporating unbiased ECs with correctly specified models can enhance statistical power. However, EC borrowing is not a free lunch, as including biased ECs may diminish power.

The power issue motivates us to develop a method that selectively incorporates unbiased ECs rather than indiscriminately borrowing all ECs. Unlike observational studies, where the assumption of no unmeasured confounders is untestable, a key advantage of hybrid controlled trials is that the bias in ECs can be identified by comparing EC units to randomized control units. Existing methods mitigate hidden bias by penalized bias estimation and selective borrowing (Gao et al., 2025), where selection consistency depends on asymptotic arguments, potentially leading to inferior performance in small samples.

We propose a novel approach called Conformal Selective Borrowing (CSB), which tests the comparability of ECs and selectively incorporates them using conformal inference (Vovk et al., 2005; Lei et al., 2018). We measure the bias of each EC using a score function that can flexibly accommodate parametric or machine-learning models. We then calibrate this score to a conformal $p$-value, which test the exchangeability of each EC. These conformal $p$-values are valid in finite samples, distribution-free, and do not depend on the asymptotic properties of models. CSB offers three advantages: (i) individual borrowing decisions for each EC, (ii) flexibility in using parametric or machine learning models for bias estimation, and (iii) finite-sample guarantees with stable performance in small samples.

In summary, the proposed methods leverage the two key advantages of hybrid controlled trials: (i) randomization within the RCT data allows us to use FRT to control the type I error rate, and (ii) the presence of randomized controls enables us to evaluate bias in ECs using conformal $p$-values, selectively borrow unbiased ECs, and enhance power. We account for selection uncertainty in FRT and offer valid post-selection inference. Both FRT and CSB are model-free, distribution-free, and maintain finite-sample exact properties, allowing them to flexibly incorporate state-of-the-art machine learning methods while remaining valid for any sample size or data distribution. To ensure robust performance across varying bias magnitudes, we propose a data-adaptive procedure for determining the selection threshold to minimize the MSE of the CSB estimator. Our MSE-guided adaptive threshold offers key advantages: (i) it improves FRT power over RCT-only analysis when EC bias is negligible or detectable; when the bias is non-negligible yet difficult to detect, it may lead to power loss, though FRT still maintains valid Type I error control; (ii) it enables CSB to serve as both a powerful test statistic and an accurate ATE estimator; (iii) the em-

pirical MSE of CSB can be approximated leveraging the RCT-only estimator, making the procedure practically feasible, and we provide a non-asymptotic excess risk bound for its performance. The advantages of our approach are shown via simulations and a lung cancer RCT with ECs from the National Cancer Database.

## 1.1. Related work

**Hybrid controlled trials** aim to integrate ECs to boost RCT efficiency (Pocock, 1976). For an overview of RCT and RWD integration, see Colnet et al. (2024). A key challenge is biases in ECs, which stem from factors like selection bias, non-concurrency, and measurement error (U.S. Food and Drug Administration, 2023). Statistically, biases are categorized as measured and unmeasured confounding. Measured confounding, or covariate shift, refers to systematic differences in observed covariates between RCs and ECs. To address measured confounding, covariate balancing techniques such as matching, inverse propensity score weighting, calibration weighting, and their augmented counterparts can be employed (Li et al., 2023b; Valancius et al., 2024; Li & Luedtke, 2023). When there is unmeasured confounding between RCT and EC, a rich body of literature addresses the hidden bias through various strategies, including test-then-pool (Viele et al., 2014; Yuan et al., 2019; Li et al., 2020; Ventz et al., 2022; Liu et al., 2022; Yang et al., 2023; Gao & Yang, 2023; Dang et al., 2023), weighted combination (Chen et al., 2020; 2021a; Cheng & Cai, 2021; Li et al., 2022; Oberst et al., 2022; Rosenman et al., 2023; Chen et al., 2023; Karlsson et al., 2024), selective borrowing (Chen et al., 2021b; Li et al., 2023a; Zhai & Han, 2022; Gao et al., 2025; Huang et al., 2023), bias modeling (Stuart & Rubin, 2008; Cheng et al., 2023; Li & Jemielita, 2023; van der Laan et al., 2024; Yang et al., 2024; Gu et al., 2024), control variates or prognostic adjustment (Yang & Ding, 2020; Guo et al., 2022; Schuler et al., 2022; Gagnon-Bartsch et al., 2023), Bayesian methods (Hobbs et al., 2011; Schmidli et al., 2014; Jiang et al., 2023; Kwiatkowski et al., 2024; Alt et al., 2024; Lin et al., 2024; 2025), and sensitivity analysis (Yi et al., 2023). None of them use randomization inference or conformal inference to address unmeasured confounding in hybrid controlled trials with a small sample size.

**Randomization inference**, introduced by Fisher (1935), provides finite-sample exact $p$-values for any test statistic and is widely endorsed (Rosenberger et al., 2019; Proschan & Dodd, 2019; Young, 2019; Bind & Rubin, 2020; Carter et al., 2023). Randomization tests are useful for small sample trials or complex designs, including cluster experiments with few clusters (Rabideau & Wang, 2021) and adaptive experiments (Simon & Simon, 2011; Plamadeala & Rosenberger, 2012; Nair & Janson, 2023; Freidling et al., 2024). Randomization tests have appeared in regulatory guidance documents to ensure type I error rate control in adaptive

designs when conventional statistical methods fail (European Medicines Agency, 2015; U.S. Food and Drug Administration, 2019; Carter et al., 2023). For an overview of randomization inference, see Zhang & Zhao (2023) and Ritzwoller et al. (2024). Nevertheless, the randomization inference hasn't been applied to hybrid controlled trials, especially with selective borrowing to address unmeasured confounding.

**Conformal inference**, or conformal prediction, is a model-free method providing finite-sample valid uncertainty quantification for individual predictions (Vovk et al., 2005), particularly useful in high-stakes scenarios with black-box machine learning models (Angelopoulos & Bates, 2023). Two main applications are most relevant to this paper. The first involves using conformal inference to infer individual treatment effects (Chernozhukov et al., 2021; Lei & Candès, 2021). The second line is in outlier detection (Guan & Tibshirani, 2022; Bates et al., 2023; Liang et al., 2024). These studies inspire us to treat biased ECs as outliers and use conformal $p$-values to test their exchangeability. Our primary goal, however, is to boost FRT power by selectively borrowing unbiased ECs with conformal $p$-values. The adaptive selection threshold that minimized the estimator's MSE is also a novel approach.

## 2. Randomization inference framework

### 2.1. Preliminaries

Consider $n_{\mathcal{R}}$ patients in the RCT, $n_{\mathcal{E}}$ patients in the EC group, and $n = n_{\mathcal{R}} + n_{\mathcal{E}}$ patients in total. Let $S = 1$ for patients in RCT and $S = 0$ for patients in the EC group. Let $A$ denote the binary treatment, where $A = 1$ stands for treatment and $A = 0$ stands for control. We denote $\mathcal{T} = \{i : A_i = 1, S_i = 1\}$, $\mathcal{C} = \{i : A_i = 0, S_i = 1\}$, $\mathcal{R} = \mathcal{T} \cup \mathcal{C}$, and $\mathcal{E} = \{i : S_i = 0\}$. Let $X$ denote the baseline covariates, $Y$ denote the observed outcome, and $Y(0)$ and $Y(1)$ denote the potential outcomes. In an RCT, we randomize $n_{\mathcal{R}}$ patients into either the treatment or control groups based on the known propensity score $e(x) = \mathbb{P}(A = 1 \mid X = x, S = 1)$. This results in $n_1$ patients in the treatment group and $n_0$ patients in the control group. For $n_{\mathcal{E}}$ patients in the EC group, since all of them are under control, we have $A = 0$ for $S = 0$. Let $\pi(x) = \mathbb{P}(S = 1 \mid X = x)$ denote the sampling score of participating in the RCT. We consider the average treatment effect in the RCT population as our estimand $\tau = \mathbb{E}\{Y(1) - Y(0) \mid S = 1\}$. For RCT data, the following standard identification assumptions are considered (Imbens & Rubin, 2015).

**Assumption 2.1** (RCT identification). (i) (Consistency) $Y = AY(1) + (1 - A)Y(0)$. (ii) (Positivity) $0 < e(x) < 1$ for all $x$ such that $f_{X|S}(x|1) > 0$, where $f_{X|S}(x|s)$ is the conditional p.d.f. of $X$ given $S = s$. (iii) (Randomization)

$Y(a) \perp\!\!\!\perp A \mid (X, S = 1), a = 0, 1$.

Under Assumption 2.1, $\tau$ is identifiable based on RCT data. We denote the conditional outcome mean functions by $\mu_a(x) = \mathbb{E}(Y \mid X = x, A = a, S = 1), a = 0, 1$. We estimate $\mu_a(x)$ and $e(x)$ with only RCT data and denote the estimated functions by $\hat{\mu}_{a,\mathcal{R}}(x)$ and $\hat{e}(x)$, respectively. An RCT-only doubly robust estimator of $\tau$ is

$$\hat{\tau}_{\mathcal{R}} = \frac{1}{n_{\mathcal{R}}} \sum_{i=1}^{n} S_i \Bigg[ \hat{\mu}_{1,\mathcal{R}}(X_i) + \frac{A_i}{\hat{e}(X_i)} \{Y_i - \hat{\mu}_{1,\mathcal{R}}(X_i)\} \\ - \hat{\mu}_{0,\mathcal{R}}(X_i) - \frac{1 - A_i}{1 - \hat{e}(X_i)} \{Y_i - \hat{\mu}_{0,\mathcal{R}}(X_i)\} \Bigg],$$

which is referred to as the No Borrowing (NB) approach hereafter. In RCTs, since the propensity score model $e(x)$ is known, $\hat{\tau}_{\mathcal{R}}$ is consistent and asymptotically normal regardless of whether $\mu_a(x)$ is correctly specified for $a = 0, 1$. Thus, $\hat{\tau}_{\mathcal{R}}$ serves as a model-assisted covariate-adjusted ATE estimator whose asymptotic variance attains the semiparametric efficiency bound if $\mu_a(x)$ is correctly specified for $a = 0, 1$. The efficiency of $\hat{\tau}_{\mathcal{R}}$ could be further improved by borrowing information from EC data. To incorporate EC data for estimating $\tau$, many scholars have considered the following assumption (Li et al., 2023b).

**Assumption 2.2** (Mean exchangeability). $\mathbb{E}\{Y(0) \mid X, S = 0\} = \mathbb{E}\{Y(0) \mid X, S = 1\}$.

Under Assumptions 2.1 and 2.2, $\tau$ could be identified with both RCT and EC data. We estimate $\mu_0(x)$ with RCT and EC data and denote the estimated functions by $\hat{\mu}_{0,\mathcal{R}+\mathcal{E}}(x)$. Let $\hat{\pi}_{\mathcal{E}}(x)$ denote the estimated sampling score. The variance ratio between randomized controls and ECs is denoted by $r(x) = \mathbb{V}\{Y(0) \mid X = x, A = 0, S = 1\}/\mathbb{V}\{Y(0) \mid X = x, A = 0, S = 0\}$. Let $\hat{r}_{\mathcal{E}}(x)$ denote the estimated variance ratio. Li et al. (2023b) proposed a doubly robust estimator of $\tau$:

$$\hat{\tau}_{\mathcal{R}+\mathcal{E}} = \frac{1}{n_{\mathcal{R}}} \sum_{i=1}^{n} \Bigg[ S_i \hat{\mu}_{1,\mathcal{R}}(X_i) + S_i \frac{A_i}{\hat{e}(X_i)} \{Y_i - \hat{\mu}_{1,\mathcal{R}}(X_i)\} \\ - S_i \hat{\mu}_{0,\mathcal{R}+\mathcal{E}}(X_i) - W_i \{Y_i - \hat{\mu}_{0,\mathcal{R}+\mathcal{E}}(X_i)\} \Bigg], \quad (1)$$

$$W_i = \hat{\pi}_{\mathcal{E}}(X_i) \frac{S_i(1 - A_i) + (1 - S_i)\hat{r}_{\mathcal{E}}(X_i)}{\hat{\pi}_{\mathcal{E}}(X_i)\{1 - \hat{e}(X_i)\} + \{1 - \hat{\pi}_{\mathcal{E}}(X_i)\}\hat{r}_{\mathcal{E}}(X_i)}.$$

$\hat{\tau}_{\mathcal{R}+\mathcal{E}}$ is referred to as the Full Borrowing (FB) approach hereafter. The term "Full" here refers to incorporating the full set of ECs to construct $\hat{\tau}_{\mathcal{R}+\mathcal{E}}$, while down-weighting those ECs based on similarity measured by $X$, thereby addressing bias caused by observed confounders. $\hat{\tau}_{\mathcal{R}+\mathcal{E}}$ is consistent and asymptotically normal if either (i) $\mu_a(x)$ is correctly specified for $a = 0, 1$, or (ii) both $\pi(x)$ and $e(x)$ are correctly specified. If all models for $\mu_a(x)$, $a = 0, 1$, $\pi(x)$, and $e(x)$ are correctly specified, $\hat{\tau}_{\mathcal{R}}$ achieves the semiparametric efficiency bound.

However, asymptotic inference for $\hat{\tau}_{\mathcal{R}+\mathcal{E}}$ may be invalid due to three main reasons: (i) it assumes $n_{\mathcal{R}} \to \infty$, which contradicts the motivation for EC borrowing, where the sample size of the RCT is typically small; (ii) it relies on the correct specification of at least one of the two nuisance models, which may be violated because sophisticated models are difficult to work with under small sample sizes; and (iii) it depends on Assumption 2.2, which may be violated due to unmeasured confounders. To address these issues, we consider a finite-sample exact randomization inference framework that maintains strict type I error rate control even if all models are misspecified and Assumption 2.2 fails. We consider $\hat{\tau}_{\mathcal{R}}$ and $\hat{\tau}_{\mathcal{R}+\mathcal{E}}$ as candidate test statistics and propose a new class of test statistics in Section 3 to achieve improved power across various scenarios.

## 2.2. Fisher randomization test

In the randomization inference framework, we are conditional on the potential outcomes $Y_i(a)$ and covariates $X_i$ for $i \in \mathcal{R} \cup \mathcal{E}$, and consider the randomized assignment $\boldsymbol{A} = (A_1, \ldots, A_n)$ as the sole source of randomness. Since $A_i$ for $i \in \mathcal{R}$ is well controlled and known in the RCT, we can leverage this advantage to guarantee the validity of inference without any additional assumptions. Let $\mathcal{A}$ denote the set of all possible assignments generated by the actual RCT design. Since all external units are under control, we have $A_i = 0$ for $i \in \mathcal{E}$. Randomization inference accommodates not only Bernoulli trials with $A_i \overset{i.i.d.}{\sim} \text{Bernoulli}(p)$ for $i \in \mathcal{R}$ but also complex designs like covariate-adaptive randomization (Rosenberger & Lachin, 2015).

Consider Fisher's sharp null hypothesis $H_0 : Y_i(0) = Y_i(1)$, $\forall i \in \mathcal{R}$, which states no treatment effect for any units in RCT. Based on $H_0$, we could impute all potential outcomes $Y_i^{\text{imp}}(0) = Y_i^{\text{imp}}(1) = Y_i$ for $i \in \mathcal{R}$. Let $T(\boldsymbol{A})$ denote the test statistic, which depends on the assignment $\boldsymbol{A} \in \mathcal{A}$. $T(\boldsymbol{A})$ could be $|\hat{\tau}_{\mathcal{R}}(\boldsymbol{A})|$, $|\hat{\tau}_{\mathcal{R}+\mathcal{E}}(\boldsymbol{A})|$, or the estimator introduced in Section 3. The theoretical guarantee of type I error rate control holds for *any test statistic*, including those involving ECs, even if these ECs have hidden biases. This is one of the key merits of randomization inference. We define the $p$-value for measuring the extremeness of the observed $T(\boldsymbol{A})$ against $H_0$ as

$$p^{\text{FRT}} = \mathbb{P}_{\boldsymbol{A}^*} \{T(\boldsymbol{A}^*) \geq T(\boldsymbol{A})\},$$

where $\boldsymbol{A}^* \in \mathcal{A}$ has the same distribution as $\boldsymbol{A}$ and is independent of $\boldsymbol{A}$, and $\mathbb{P}_{\boldsymbol{A}^*}$ is taken over the distribution of $\boldsymbol{A}^*$.

**Theorem 2.3.** *Under $H_0$, for $\alpha \in (0,1)$, we have $\mathbb{P}_{\boldsymbol{A}}(p^{\text{FRT}} \leq \alpha) \leq \alpha$, where $\mathbb{P}_{\boldsymbol{A}}$ is taken over the distribution of $\boldsymbol{A}$. If we further assume that $T(\boldsymbol{A})$ takes distinct values for different $\boldsymbol{A} \in \mathcal{A}$, then we have $\mathbb{P}_{\boldsymbol{A}}(p^{\text{FRT}} \leq \alpha) = \lfloor \alpha |\mathcal{A}| \rfloor / |\mathcal{A}| > \alpha - 1/|\mathcal{A}|$, where $\lfloor x \rfloor$ represents the greatest integer less than or equal to $x$.*

In practice, we use Monte Carlo to approximate $p^{\text{FRT}}$. Based on the RCT's actual randomization, we generate the new assignment $A_i^b$ for $i \in \mathcal{R}$ and set $A_i^b \equiv 0$ for $i \in \mathcal{E}$ since the randomization in the RCT does not affect the assignments of the ECs. A caveat is that the assignment of ECs should not be permuted, as this would violate the "analyze as you randomize" principle and compromise the validity of the FRT. The new assignment vector is denoted as $\boldsymbol{A}^b = (A_1^b, \ldots, A_n^b)$. We generate assignments for $B$ times and obtain $\hat{p}^{\text{FRT}} = \left[ \sum_{b=1}^{B} \mathbb{I}\{T(\boldsymbol{A}^b) \geq T(\boldsymbol{A})\} + 1 \right]/(B+1)$, where the "+1" term accounts for $\boldsymbol{A}$ itself.

Theorem 2.3 shows that FRT exactly controls the type I error rate in finite samples, regardless of Assumption 2.2, because, under $H_0$, the reference distribution is derived from true randomization, which is well-controlled in clinical trials. However, the power of FRT heavily depends on the choice of test statistic, making it the most critical decision in randomization inference.

## 2.3. Model-based power analysis

There are two primary approaches for conducting a power analysis of FRT: model-based or simulation-based (Rosenberger & Lachin, 2015). We first perform a model-based power analysis under Assumption 2.2, highlighting how low variance of a consistent test statistic enhances the power of FRT. In the following section, we conduct a simulation-based power analysis for a more challenging scenario where Assumption 2.2 does not hold, showing that the bias of an inconsistent test statistic reduces the power of FRT.

Let $M$ denote the total number of possible assignments, $F_{1,n,M}(t) = \mathbb{P}_{\boldsymbol{A}}(T(\boldsymbol{A}) \leq t)$ denote the randomization distribution of $T(\boldsymbol{A})$, and $F_{0,n,M}(t) = \mathbb{P}_{\boldsymbol{A}^*}(T(\boldsymbol{A}^*) \leq t)$ denote the reference distribution of $T(\boldsymbol{A}^*)$ under $H_0$. Both $F_{1,n,M}$ and $F_{0,n,M}$ are discrete in finite samples. To apply empirical process theory and derive asymptotic rates for testing power, we assume continuous super-population distributions $F_{1,n}$ and $F_{0,n}$, with $F_{1,n,M}$ and $F_{0,n,M}$ representing the empirical distribution functions based on $M$ independent samples drawn from $F_{1,n}$ and $F_{0,n}$, respectively. In cases where these assumptions do not hold, FRT still controls the type I error rate, and we will investigate its power through simulation in Section 4. Based on those notations, the $p$-value and the power can be expressed as $p^{\text{FRT}} = \mathbb{P}_{\boldsymbol{A}^*} \{T(\boldsymbol{A}^*) \geq T(\boldsymbol{A})\} = 1 - F_{0,n,M}(T(\boldsymbol{A}))$, and $\psi_{n,M} = \mathbb{P}_{\boldsymbol{A}}(p^{\text{FRT}} \leq \alpha) = \mathbb{P}_{\boldsymbol{A}}\{1 - F_{0,n,M}(T(\boldsymbol{A})) \leq \alpha\} = 1 - F_{1,n,M}(F_{0,n,M}^{-1}(1-\alpha))$.

**Theorem 2.4.** *For fixed $n > 0$, suppose*

*(a) There are continuous cumulative distribution functions (c.d.f.) $F_{0,n}$ and $F_{1,n}$, such that $F_{0,n,M}$ and $F_{1,n,M}$ are the empirical distribution functions based on $M$ independent samples drawn from $F_{0,n}$ and $F_{1,n}$, respectively.*

*(b) There is $\sigma_n > 0$ and a continuous c.d.f. $F$ such that $F_{0,n}(t) = F(t/\sigma_n)$ for all $t \in \mathbb{R}$.*

*(c) For ATE $\tau$, $F_{1,n}(t) = F_{0,n}(t - \tau) = F((t - \tau)/\sigma_n)$, for all $t \in \mathbb{R}$.*

*For $0 < \iota < 0.5$ and sufficiently large $M$,*

$$\mathbb{E}(\psi_{n,M}) \geq 1 - F(F^{-1}(1-\alpha) - \tau/\sigma_n) - O(M^{-0.5+\iota}),$$

*where $\mathbb{E}$ is over $M$ independent samples from $F_{0,n}$ and $F_{1,n}$.*

For a given $\tau \neq 0$, significance level $\alpha$, and design with possible assignments $M$, Theorem 2.4 shows that the power of the FRT also depends on the variance of the test statistic, $\sigma_n$. Under Assumptions 2.1 and 2.2, and with all working models correctly specified, $\hat{\tau}_{\mathcal{R}+\mathcal{E}}$ is consistent and has a variance that is less than or equal to that of $\hat{\tau}_\mathcal{R}$ (Li et al., 2023b). Thus, when there is no hidden bias, using $\hat{\tau}_{\mathcal{R}+\mathcal{E}}$ as the test statistic improves the power of the FRT compared to $\hat{\tau}_\mathcal{R}$, as shown in subplot (B) of Figure 1.

### 2.4. Simulation-based power analysis

When unmeasured confounding exists between RCT and EC data, Assumption 2.2 is violated, rendering $\hat{\tau}_{\mathcal{R}+\mathcal{E}}$ inconsistent. In such cases, asymptotic inference based on $\hat{\tau}_{\mathcal{R}+\mathcal{E}}$ is invalid and fails to control the type I error rate. In contrast, since Theorem 2.3 holds for any test statistic, FRT can still control the type I error with the inconsistent test statistic $\hat{\tau}_{\mathcal{R}+\mathcal{E}}$, highlighting a core merit of FRTs. However, the violation of Assumption 2.2 subsequently causes Assumption (c) in Theorem 2.4 to be unfulfilled, rendering FRT with $\hat{\tau}_{\mathcal{R}+\mathcal{E}}$ unable to achieve a power improvement over FRT with $\hat{\tau}_\mathcal{R}$. Furthermore, employing $\hat{\tau}_{\mathcal{R}+\mathcal{E}}$ as the test statistic results in a substantial loss of power compared to using $\hat{\tau}_\mathcal{R}$, as illustrated in subplot (D) of Figure 1.

The trade-off between $\hat{\tau}_\mathcal{R}$ and $\hat{\tau}_{\mathcal{R}+\mathcal{E}}$ generally arises between a causal estimator that ignores additional information and assumptions and one that incorporates them but risks bias if the assumptions fail (Rothenhäusler, 2020; Rothenhäusler et al., 2021). In the next section, instead of choosing between $\hat{\tau}_\mathcal{R}$ and $\hat{\tau}_{\mathcal{R}+\mathcal{E}}$, we construct a class of ATE estimators, $\hat{\tau}_\gamma$, indexed by a tuning parameter $\gamma$ and encompassing $\hat{\tau}_\mathcal{R}$ and $\hat{\tau}_{\mathcal{R}+\mathcal{E}}$ as special cases. We then propose a data-adaptive procedure to select $\gamma$ that minimizes the MSE of $\hat{\tau}_\gamma$, thereby enhancing the power of FRT by using $\hat{\tau}_\gamma$ as the test statistic.

## 3. Conformal Selective Borrowing

### 3.1. A class of estimators

Motivated by heterogeneous scenarios where some ECs satisfy Assumption 2.2 while others do not, we propose an individualized test-then-pool approach that leverages conformal inference to select comparable ECs. The conformal $p$-value $p_j^* \in (0, 1]$ is used to test the exchangeability of each EC $j \in \mathcal{E}$. The selected EC set is then defined as $\hat{\mathcal{E}}(\gamma) = \{j \in \mathcal{E} : p_j^* > \gamma\}$, where $\gamma \in [0, 1]$ is a selection threshold. Substituting $\mathcal{E}$ with $\hat{\mathcal{E}}(\gamma)$ in (1), we obtain the Conformal Selective Borrowing (CSB) estimator:

$$\hat{\tau}_\gamma = \frac{1}{n_\mathcal{R}} \sum_{i=1}^n \left[ S_i \, \hat{\mu}_{1,\mathcal{R}}(X_i) + S_i \frac{A_i}{\hat{e}(X_i)} \{Y_i - \hat{\mu}_{1,\mathcal{R}}(X_i)\} \right.$$
$$\left. - S_i \, \hat{\mu}_{0,\mathcal{R}+\hat{\mathcal{E}}(\gamma)}(X_i) - V_i \{Y_i - \hat{\mu}_{0,\mathcal{R}+\hat{\mathcal{E}}(\gamma)}(X_i)\} \right], \quad (2)$$

$$V_i = \hat{\pi}_{\hat{\mathcal{E}}(\gamma)}(X_i)$$
$$\times \frac{S_i(1 - A_i) + (1 - S_i)\mathbb{I}\{i \in \hat{\mathcal{E}}(\gamma)\}\hat{r}_{\hat{\mathcal{E}}(\gamma)}(X_i)}{\hat{\pi}_{\hat{\mathcal{E}}(\gamma)}(X_i)\{1 - \hat{e}(X_i)\} + \{1 - \hat{\pi}_{\hat{\mathcal{E}}(\gamma)}(X_i)\}\hat{r}_{\hat{\mathcal{E}}(\gamma)}(X_i)}.$$

CSB represents a class of ATE estimators: when $\gamma = 1$, no ECs borrowed and $\hat{\mathcal{E}}(1) = \varnothing$, we have $\hat{\tau}_1 \equiv \hat{\tau}_\mathcal{R}$; when $\gamma = 0$, all ECs borrowed and $\hat{\mathcal{E}}(0) = \mathcal{E}$, we have $\hat{\tau}_0 = \hat{\tau}_{\mathcal{R}+\mathcal{E}}$. For $0 < \gamma < 1$, $\hat{\tau}_\gamma$ balances the trade-off between borrowing more ECs with a smaller $\gamma$ and discarding more ECs with a larger $\gamma$. By using $T(\boldsymbol{A}) = |\hat{\tau}_\gamma|$ as the test statistic for FRT and allowing $\hat{\mathcal{E}}(\gamma)$ to vary with resampling $\boldsymbol{A}$ in FRT could account for selection uncertainty and provide valid post-selection inference. The following sections introduce various conformal $p$-values and a data-adaptive procedure for selecting $\gamma$ to minimize the MSE of $\hat{\tau}_\gamma$.

### 3.2. Conformal $p$-value

**Split conformal $p$-value.** We first consider split conformal inference (Papadopoulos et al., 2002). We randomly split $\mathcal{C}$ into a calibration set $\mathcal{C}_1$ and a training set $\mathcal{C} \setminus \mathcal{C}_1$ according to a prespecified sample size ratio, for example, $1 : 3$. We use a score function $s(x, y)$ to measure the "nonconformity" of $(x, y)$. For example, we can use the absolute residual as the score function: $s_i = |Y_i - \hat{f}_{-\mathcal{C}_1}(X_i)|$ for $i \in \mathcal{C}_1$ and $s_j = |Y_j - \hat{f}_{-\mathcal{C}_1}(X_j)|$, where $\hat{f}_{-\mathcal{C}_1}(x)$ is a prediction model fitted by the training set $\mathcal{C} \setminus \mathcal{C}_1$. Intuitively, if $(X_j, Y_j)$ is not exchangeable (see Remark 3.2 for a formal definition) with $\{(X_i, Y_i)\}_{i \in \mathcal{C}_1}$, $s_j$ should be large compared to $\{s_i\}_{i \in \mathcal{C}_1}$. Thus, we define the split conformal $p$-value as the proportion of $\{s_i\}_{i \in \mathcal{C}_1}$ that are larger than $s_j$, that is, $p_j^{\text{split}} = \{\sum_{i \in \mathcal{C}_1} \mathbb{I}(s_i \geq s_j) + 1\}/(|\mathcal{C}_1| + 1)$, where $\mathbb{I}$ is the indicator function, and the "+1" accounts for including $s_j$ itself. If $p_j^{\text{split}}$ is smaller than a threshold $\gamma$, we reject the hypothesis of exchangeability and discard EC $j$. The following theoretical guarantee states that if EC $j$ is exchangeable, the rejection rate is less than $\gamma$.

**Proposition 3.1.** *For $j \in \mathcal{E}$, suppose that $(X_j, Y_j)$ and $\{(X_i, Y_i)\}_{i \in \mathcal{C}}$ are exchangeable. For $\gamma \in (0, 1)$, we have*

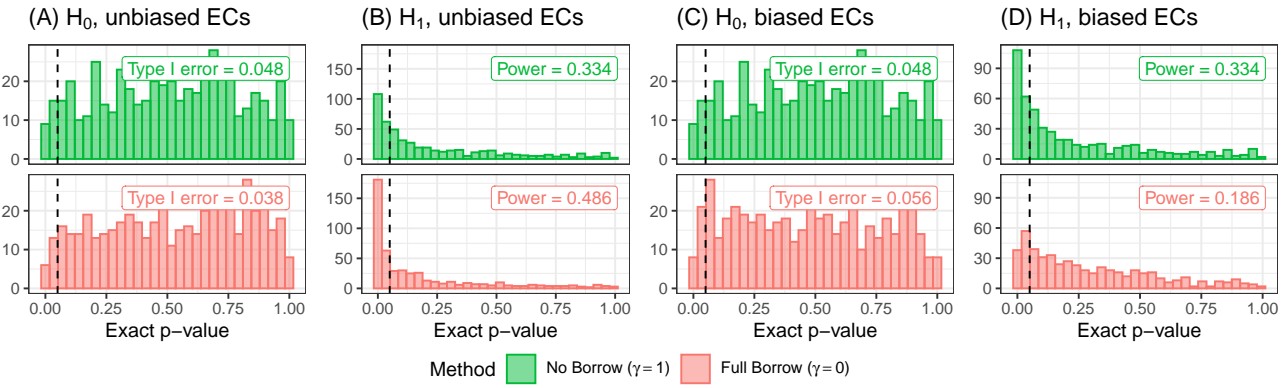

Figure 1. Simulated distributions of $p$-values under $H_0$ and $H_1$.

$\mathbb{P}(p_j^{\text{split}} \leq \gamma) \leq \gamma$. If $s_j$ and $\{s_i\}_{i \in \mathcal{C}}$ have distinct values, we have $\mathbb{P}(p_j^{\text{split}} \leq \gamma) = \{\lfloor \gamma(|\mathcal{C}_1| + 1) \rfloor\} / (|\mathcal{C}_1| + 1) > \gamma - 1/(|\mathcal{C}_1| + 1)$.

*Remark* 3.2 (Definition of exchangeability). The random variables $z_1, \ldots, z_n$ are exchangeable if, for any permutation $\omega$ of $1, \ldots, n$, the random variables $z_{\omega(1)}, \ldots, z_{\omega(n)}$ have the same joint distribution as $z_1, \ldots, z_n$. The i.i.d. assumption is stronger than exchangeability, as the latter can hold with dependence (Shafer & Vovk, 2008). The exchangeability required by conformal inference is stronger than the mean exchangeability (Assumption 2.2), which allows the construction of a statistically valid estimator within the asymptotic inference framework.

**CV+ $p$-value.** While split conformal $p$-values are computationally efficient, they lose statistical efficiency due to data splitting. CV+ (Barber et al., 2021) fully utilize training data and remain computationally feasible. We randomly split $\mathcal{C}$ into $K$ disjoint folds: $\mathcal{C} = \cup_{k=1}^{K} \mathcal{C}_k$. We use the training set $\mathcal{C} \setminus \mathcal{C}_k$ to fit prediction models $\hat{f}_{-\mathcal{C}_k}(x)$ and use the absolute residual as the score function: $s_i = |Y_i - \hat{f}_{-\mathcal{C}_{k(i)}}(X_i)|$ and $s_j^{(i)} = |Y_j - \hat{f}_{-\mathcal{C}_{k(i)}}(X_j)|$ for $i \in \mathcal{C}$, where $k(i) \in \{1, \ldots, K\}$ is a function that indicates $i \in \mathcal{C}_k$. Thus, for $i \neq i'$ and $k(i) = k(i')$, we have $s_j^{(i)} = s_j^{(i')}$. We define the CV+ $p$-value as the proportion of $\{s_i\}_{i \in \mathcal{C}}$ that are larger than the corresponding $\{s_j^{(i)}\}_{i \in \mathcal{C}}$, that is, $p_j^{\text{cv+}} = \{\sum_{i \in \mathcal{C}} \mathbb{I}(s_i \geq s_j^{(i)}) + 1\}/(|\mathcal{C}| + 1)$.

**Proposition 3.3.** *For $j \in \mathcal{E}$, suppose that $(X_j, Y_j)$ and $\{(X_i, Y_i)\}_{i \in \mathcal{C}}$ are exchangeable. For $\gamma \in (0, 1)$, we have $\mathbb{P}(p_j^{\text{cv+}} \leq \gamma) \leq 2\gamma + \{(1 - 2\gamma)(m - 1) - 1\} / (|\mathcal{C}| + m) < 2\gamma + (1 - K/|\mathcal{C}|) / (K + 1)$, where $m = |\mathcal{C}|/K$ is assumed to be an integer for simplicity.*

### 3.3. Adaptive selection threshold

Since we construct $p_j^*$ individually and make borrowing decisions collectively, one might consider choosing a selection

threshold $\gamma$ that controls the family-wise type I error rate or false discovery rate for testing the exchangeability of all ECs (Bates et al., 2023). However, in our context, the power of the conformal tests is of greater concern. The classical test-then-pool approach has been criticized for its low power in detecting hidden bias, especially with small randomized control sample sizes (Li et al., 2020). Even with effective control of the family-wise type I error rate, low-power conformal tests can allow many biased ECs to be incorrectly borrowed, increasing the MSE of $\hat{\tau}_\gamma$ and reducing the power of the FRT. Therefore, we propose a data-adaptive procedure to directly minimize the MSE of $\hat{\tau}_\gamma$.

We decompose $\text{MSE}(\gamma) \equiv \mathbb{E}(\hat{\tau}_\gamma - \tau)^2 = \{\mathbb{E}(\hat{\tau}_\gamma) - \tau\}^2 + \mathbb{V}(\hat{\tau}_\gamma)$. The main challenge lies in estimating the squared bias $\{\mathbb{E}(\hat{\tau}_\gamma - \tau)\}^2$ as the true $\tau$ is unknown. Fortunately, since the NB estimator $\hat{\tau}_1$ is consistent for $\tau$, we approximate $\{\mathbb{E}(\hat{\tau}_\gamma - \tau)\}^2$ by $\{\mathbb{E}(\hat{\tau}_\gamma - \hat{\tau}_1)\}^2 = \mathbb{E}(\hat{\tau}_\gamma - \hat{\tau}_1)^2 - \mathbb{V}(\hat{\tau}_\gamma - \hat{\tau}_1)$. We then use $(\hat{\tau}_\gamma - \hat{\tau}_1)^2$ to estimate $\mathbb{E}(\hat{\tau}_\gamma - \hat{\tau}_1)^2$ and apply bootstrap to estimate $\mathbb{V}(\hat{\tau}_\gamma)$ and $\mathbb{V}(\hat{\tau}_\gamma - \hat{\tau}_1)$. Combining these provides the estimated MSE for each $\gamma$ over finite grids, and we select the $\gamma$ that minimizes it. The complete procedure is detailed in Algorithm 1.

---

**Algorithm 1:** Adaptive Selection Threshold

**Input:** Grid $\Gamma = \{0, 0.1, \ldots, 1\}$; bootstrap times $L$.

**for** $\gamma \in \Gamma$ **do**
  Compute $\hat{\tau}_\gamma$ from the original sample.
  **for** $l = 1, \ldots, L$ **do**
    Compute $\hat{\tau}_\gamma^{(l)}$ from the $l$-th bootstrap sample.

**for** $\gamma \in \Gamma$ **do**
  Compute $\widehat{\mathbb{V}}(\hat{\tau}_\gamma - \hat{\tau}_1)$ using $\hat{\tau}_\gamma^{(l)} - \hat{\tau}_1^{(l)}$.
  Compute $\widehat{\mathbb{V}}(\hat{\tau}_\gamma)$ using $\hat{\tau}_\gamma^{(l)}$.
  $\widehat{\text{MSE}}(\gamma) = (\hat{\tau}_\gamma - \hat{\tau}_1)^2 - \widehat{\mathbb{V}}(\hat{\tau}_\gamma - \hat{\tau}_1) + \widehat{\mathbb{V}}(\hat{\tau}_\gamma)$.

**Output:** $\hat{\gamma} = \arg\min_{\gamma \in \Gamma} \widehat{\text{MSE}}(\gamma)$

---

We theoretically analyze the procedure from a non-asymptotic perspective (Wainwright, 2019). Decomposing $\hat{\tau}_\gamma = \tau + \delta_\gamma + \epsilon_\gamma$, where $\delta_\gamma \equiv \mathbb{E}(\hat{\tau}_\gamma) - \tau$ and $\mathbb{E}(\epsilon_\gamma) = 0$. Let $\kappa_\gamma^2 \equiv \mathbb{V}(\hat{\tau}_\gamma - \hat{\tau}_1) = \mathbb{V}(\epsilon_\gamma - \epsilon_1)$ and $\sigma_\gamma^2 \equiv \mathbb{V}(\hat{\tau}_\gamma) = \mathbb{V}(\epsilon_\gamma)$.

**Theorem 3.4.** *For fixed $n > 0$ and $\gamma \in \Gamma$, let $\epsilon_\gamma$ be a centered sub-Gaussian variable with parameter $\phi_\gamma > 0$, i.e., $\mathbb{E}\exp(\lambda\epsilon_\gamma) \leq \exp(\phi_\gamma^2\lambda^2/2)$ for all $\lambda \in \mathbb{R}$. For $\iota > 0$, there exists $c > 0$ such that with probability at least $1 - 4\iota$:*

$$\max_{\gamma \in \Gamma}\left|\widehat{\mathrm{MSE}}(\gamma) - \mathrm{MSE}(\gamma)\right| \leq c\Delta|\delta_1| + c\Delta\Phi\sqrt{\log(|\Gamma|/\iota)}$$
$$+ \max\left\{c\Phi^2\sqrt{\log(|\Gamma|/\iota)},\ c\Phi^2\log(|\Gamma|/\iota)\right\}$$
$$+ \max_{\gamma \in \Gamma}|\widehat{\mathbb{V}}(\hat{\tau}_\gamma - \hat{\tau}_1) - \kappa_\gamma^2| + \max_{\gamma \in \Gamma}|\widehat{\mathbb{V}}(\hat{\tau}_\gamma) - \sigma_\gamma^2|,$$

*where $\Delta = \max_{\gamma \in \Gamma}|\delta_\gamma|$, $\Phi = \max_{\gamma \in \Gamma}\phi_\gamma$, $\delta_1$ is the bias of $\hat{\tau}_1$, and $|\Gamma|$ is the cardinality of $\Gamma$.*

Theorem 3.4 shows that the discrepancy between the estimated and true MSE vanishes if $\Delta$ is bounded and the bias of the consistent estimator $\hat{\tau}_1$, the maximum standard deviation proxy $\Phi$, and the variance estimation errors are sufficiently small.

**Theorem 3.5.** *Under the same assumptions as in Theorem 3.4, for any $\iota > 0$, there exists a constant $c > 0$ such that, with probability at least $1 - 8\iota$, the following holds:*

$$(\hat{\tau}_{\hat{\gamma}} - \tau)^2 - \min_{\gamma \in \Gamma}(\hat{\tau}_\gamma - \tau)^2 \leq 2c\Delta|\delta_1| + 2c\Delta\Phi\sqrt{\log(|\Gamma|/\iota)}$$
$$+ 2\max\left\{c\Phi^2\sqrt{\log(|\Gamma|/\iota)},\ c\Phi^2\log(|\Gamma|/\iota)\right\}$$
$$+ 2\max_{\gamma \in \Gamma}|\widehat{\mathbb{V}}(\hat{\tau}_\gamma - \hat{\tau}_1) - \kappa_\gamma^2| + 2\max_{\gamma \in \Gamma}|\widehat{\mathbb{V}}(\hat{\tau}_\gamma) - \sigma_\gamma^2|.$$

Theorem 3.5 provides a bound for the excess risk of $\hat{\tau}_{\hat{\gamma}}$ in comparison to the oracle estimator. Although $\hat{\tau}_{\hat{\gamma}}$ generally outperforms $\hat{\tau}_1$ in terms of MSE, it may exhibit excess risk in certain challenging cases, as shown in Figure 2 (C) in the simulation. This phenomenon highlights that $\hat{\tau}_{\hat{\gamma}}$ behaves similarly to the Hodges estimator (Le Cam, 1953) and to integrated estimators in data fusion (Yang et al., 2023; Oberst et al., 2022): improving upon the baseline estimator (here, the No Borrow estimator) in certain regions of the parameter space (where there is no bias in ECs) inevitably leads to worse performance in other regions (where the bias in ECs is difficult to detect). FRT still controls the type I error rate even if excess risk is present or the assumptions in Theorem 3.4 are not satisfied.

## 4. Simulation

We conduct simulations to evaluate the repeated sampling performance of the proposed methods under small sample sizes and varying magnitudes of hidden bias, including challenging cases where separating biased ECs is difficult. Specifically, the sample sizes for the randomized treatment, randomized control, and EC groups are set as $(n_1, n_0, n_{\mathcal{E}}) = (50, 25, 50)$. Similar results for a larger EC sample size ($n_{\mathcal{E}} = 300$) are included in the Appendix. We generate covariates $X \sim \mathrm{Unif}(-2, 2)$ with dimension $p = 2$. The sampling indicator $S \sim \mathrm{Bernoulli}(\pi(X))$ is generated with $\pi(X) = \{1 + \exp(\eta_0 + X^{\mathrm{T}}\eta)\}^{-1}$, where $\eta_0$ is chosen to ensure $\mathbb{E}(S) = n_{\mathcal{R}}/n$, and $\eta = (0.1, 0.1)$. The assignment is generated by $A \sim \mathrm{Bernoulli}(n_1/n_{\mathcal{R}})$ for $S = 1$ and $A = 0$ for $S = 0$. Let $\varepsilon \sim N(0, 1)$ denote the noise. For the RCT sample ($S = 1$), we generate the potential outcomes as $Y(0) = X^{\mathrm{T}}\beta_0 + \varepsilon$ with $\beta_0 = (1, 1)$, and $Y(1) = \tau_0 + X^{\mathrm{T}}\beta_1 + \varepsilon$ with $\tau_0 = 0.4$ and $\beta_1 = (2, 2)$. For the EC sample ($S = 0$), we consider two scenarios: (i) the scenario without hidden bias, where $Y(0) = X^{\mathrm{T}}\beta_0 + 0.5\varepsilon$; (ii) the scenario where part of the ECs have hidden bias $b$, where a random proportion $\rho$ of the ECs is biased, with $Y(0) = -b + X^{\mathrm{T}}\beta_0 + 0.5\varepsilon$, and the remaining proportion $(1 - \rho)$ are unbiased, with $Y(0) = X^{\mathrm{T}}\beta_0 + 0.5\varepsilon$. We consider proportions of biased ECs $\rho = 50\%$ and magnitudes of hidden bias $b = 1, 2, \ldots, 8$. Note that hidden bias refers to bias that remains due to unmeasured confounders, even after balancing the observed covariates. Under the alternative hypothesis, the observed outcome is $Y = AY(1) + (1 - A)Y(0)$; under the sharp null hypothesis, the observed outcome is $Y = Y(0)$. We consider NB, FB, and CSB with the adaptive selection threshold as estimators of $\tau$ and test statistics for FRT. We also consider Adaptive Lasso Selective Borrowing (ALSB) by Gao et al. (2025). Given its higher computational cost (approximately 10 times slower than CSB), we omit FRTs for this method and instead compare CSB+FRT with ALSB+asymptotic inference in the Appendix. CV+ $p$-values are used with 10 folds. We set $B = 5000$ to approximate $p^{\mathrm{FRT}}$ and replicate the simulation 500 times per scenario.

Figure 2 displays performance metrics for $b = 0, 1, \ldots, 8$. In the first case ($b = 0$): (i) all methods exhibit negligible bias; (ii) FB and CSB reduce MSE by 42% and 20%, respectively, compared to NB; (iii) all methods effectively control the type I error rate; and (iv) FB and CSB increase power by 46% and 45%, respectively, compared to NB. In the following eight cases ($b = 1, \ldots, 8$): (i) FB exhibits a large bias, approximately 125%-203% of its standard deviation (SD). The absolute bias of FB decreases with $b$ when $b \geq 3$ because large $b$ values increase $\mathbb{V}\{Y(0) \mid X = x, A = 0, S = 0\}$, causing FB to downweight ECs with small $\hat{r}(X_i)$ in (1). CSB performs better at bias control, with bias ranging from 0%-22% of its SD; (ii) compared to NB, FB increases MSE by up to 454%, and CSB decreases MSE by 13%-16% (except when $b = 1, 2$, where MSE increases by 1%-18%). (iii) In line with Theorem 2.3, all methods control the type I error rate well; (iv) compared to NB, FB decreases power by up to 51%. In contrast, CSB increases power by 13%-36% (except when $b = 2, 3, 4$, where

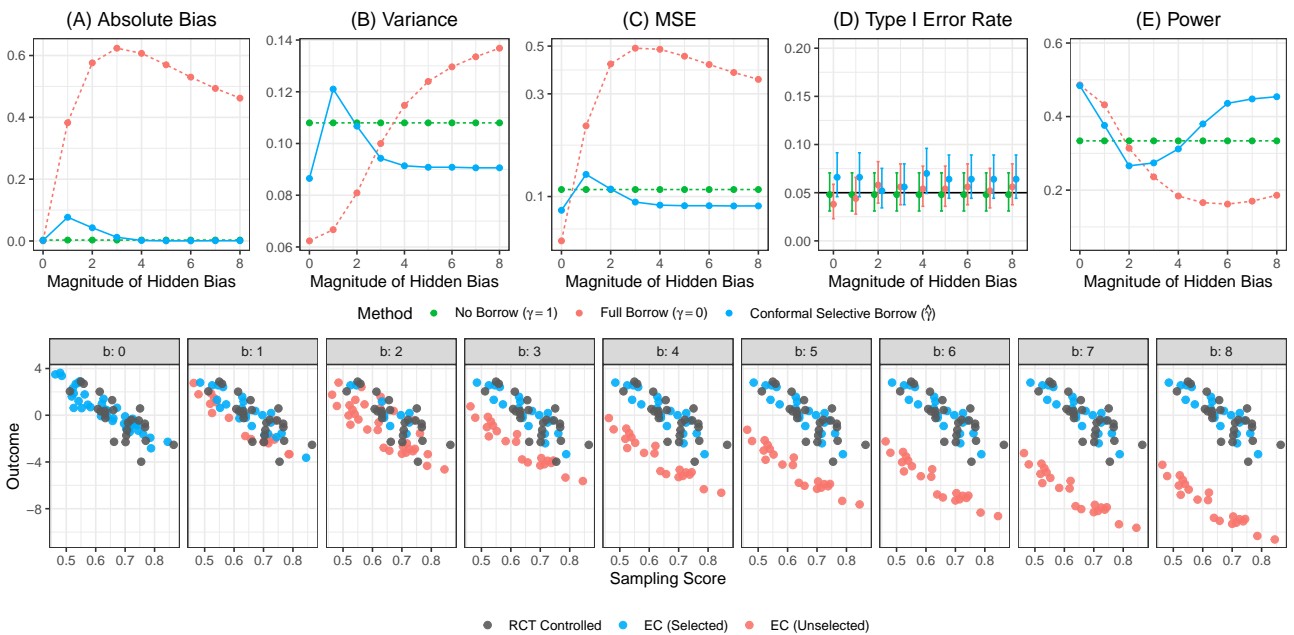

*Figure 2.* Simulation results across different hidden bias magnitudes $b$.

power decreases by 7%-20%). In challenging cases where $0 < b \leq 4$, the efficiency loss of CSB occurs because small biases make it hard to distinguish biased ECs from unbiased ECs. Such loss is inevitable when aiming to gain efficiency in scenarios without hidden bias, a phenomenon known in the transfer learning literature as the cost of transferability detection (Cai et al., 2024). This phenomenon also occurs for other data integration estimators under hidden bias (see Figure 2 in Yang et al. (2023), Figure 4 in Oberst et al. (2022), and Figure 2 in Lin et al. (2024)). Finally, we examine the selection performance of CSB. We do not expect CSB to perfectly separate biased ECs from unbiased ones due to (i) the small sample size of randomized controls and (ii) finite sample noise. As shown in Figure 2, CSB discards biased ECs and some unbiased ones that aren't sufficiently similar to randomized controls, demonstrating satisfactory selection performance.

## 5. Real data application

**The CALGB 9633 and NCDB data.** We apply the proposed methods to an RCT conducted by the Cancer and Leukemia Group B (CALGB), known as CALGB 9633, which investigated the treatment effect of adjuvant chemotherapy in patients with stage IB non-small-cell lung cancer (Strauss et al., 2008). In CALGB 9633 ($S = 1$), $n_1 = 167$ patients were randomized to adjuvant chemotherapy ($A = 1$), and $n_0 = 168$ were randomized to observation ($A = 0$). We extract data for 11,700 patients from the National Cancer Database (NCDB) as the EC sample

($A = 0, S = 0$) to improve CALGB 9633's statistical efficiency. The NCDB is a clinical oncology database sourced from hospital registry data, jointly run by the American Cancer Society and the American College of Surgeons, covering 70% of U.S. cancer cases.

**RMST and pseudo-observations** We use the *Restricted Mean Survival Time* (RMST), $Y = \min(T, t^*)$, as the primary endpoint, where $T$ represents the survival time and $t^*$ is the truncation time. RMST measures survival time up to a clinically relevant truncation point and serves as a compelling alternative to the hazard ratio when the proportional hazards assumption is violated (Hernán, 2010). We consider the difference in 3-year RMST between the treatment and control groups for the RCT population $\tau = E\{Y(1) - Y(0) \mid S = 1\}$ as the estimand, where $Y(a) = \min\{T(a), 3\}$ and $T(a)$ is the potential survival time, $a = 0, 1$. Five baseline covariates in CALGB 9633 and NCDB are considered: sex, age, race, histology, and tumor size. The censoring rates of $T$ in CALGB 9633 and NCDB are 42% and 48%, respectively. We use a "once-for-all" approach to transform right-censored survival times into *pseudo-observations* for RMST, allowing standard causal inference methods as if outcomes were non-censored (Andersen et al., 2003; Overgaard et al., 2017). To address covariate-dependent censoring, we stratified by sex, race, and histology, applying transformations separately within each dataset (Andersen & Pohar Perme, 2010). The stratified Kaplan–Meier estimator is used to estimate survival functions, with pseudo-observations generated via the jack-

*Table 1.* Analysis results for CALGB 9633 + NCDB.

| Method | Est | SE | CI | Asym $p$ | Exact $p$ | #EC |
|---|---|---|---|---|---|---|
| No Borrow (Dif-in-Means) | 0.135 | 0.072 | (-0.007, 0.276) | 0.062 | 0.060 | 0 |
| No Borrow (AIPW) | 0.142 | 0.074 | (-0.003, 0.286) | 0.055 | 0.051 | 0 |
| Full Borrow | 0.241 | 0.061 | (0.122, 0.361) | <0.001 | 0.031 | 335 |
| Conformal Selective Borrow | 0.138 | 0.058 | (0.024, 0.252) | 0.018 | 0.046 | 264 |

"Est" is the estimate. "SE", "CI", and "Asym $p$" are the asymptotic standard error, confidence interval, and $p$-value, respectively. "Exact $p$" is the exact $p$-value. "#EC" is the number of borrowed ECs.

knife method, as implemented in the R package `eventglm` (Sachs & Gabriel, 2022). We treat the pseudo-observations for 3-year RMST as the outcome hereafter. More details about the real data are provided in Appendix Section D.

**Data analysis.** We apply `NB`, `FB`, and `CSB` to estimate the ATE and perform FRTs. For comparison, we also apply `NB` without covariate adjustment, i.e., difference-in-means estimator. In addition to the proposed exact $p$-value, we also compute the standard error, confidence interval, and $p$-value based on asymptotic inference for all approaches (Li et al., 2023b). Since the outcome shows a high proportion of truncation at 3 years, resulting in a highly skewed distribution, we apply the conformal quantile regression (Romano et al., 2019) to compute the conformal score. We use the Jackknife+ $p$-value (Barber et al., 2021) to achieve a better balance between statistical and computational efficiency. Table 1 presents the analysis results. For `NB` using Dif-in-Means and AIPW, asymptotic and exact $p$-values range from 0.051 to 0.062. In contrast, `FB` (using all 335 ECs) gives an asymptotic $p$-value of $< 0.001$ and an exact $p$-value of 0.031, indicating a significantly positive ATE. Similarly, `CSB` (using 178 ECs) shows an asymptotic $p$-value of 0.018 and an exact $p$-value of 0.046, also indicating a significantly positive ATE. The ATE estimate from `CSB` falls between `NB` and `FB`, indicating a trade-off between these two approaches.

## 6. Discussion

This paper proposes using FRT in hybrid controlled trials and introduces `CSB` for selectively incorporating comparable ECs, mitigating hidden bias. FRT with `CSB` maintains type I error control and improves power compared to RCT-only analysis. The proposed `CSB` estimator with an adaptive selection threshold enhances efficiency over the `NB` approach.

One limitation of our procedure is that, when the bias is non-negligible yet difficult to detect, it may incur some power loss, though it still maintains valid Type I error control. This no-free-lunch limitation is acknowledged in existing papers (Oberst et al., 2022; Lin et al., 2024), which point out that without assuming mean exchangeability of ECs, no

method can uniformly and significantly outperform RCT-only analysis across varying levels of hidden bias, although different approaches optimize the risk-reward trade-off from different perspectives. The most challenging scenarios are those where bias is non-negligible but complex to correct or difficult to detect. Our key distinctions from existing literature are twofold: (i) we prioritize exact Type I error control in small samples before seeking power gains; (ii) we optimize the risk-reward trade-off between no borrowing and full borrowing through conformal selective borrowing, motivated by real data in which some ECs are unbiased while others are not.

Heterogeneity among data sources is common in integration and transfer learning, often leading to bias or efficiency loss even after balancing measured confounders. While penalized bias estimation is a common solution, our work demonstrates that conformal inference provides greater stability and flexibility in finite samples. Extending this approach to tasks like developing individual treatment regimes (Chu et al., 2023), exploring treatment effect heterogeneity (Wu & Yang, 2022), and improving experimental design (Ruan et al., 2024) shows great potential.

Beyond the sharp null, FRTs can test the weak null asymptotically using studentized or prepivoted statistics (Wu & Ding, 2021; Cohen & Fogarty, 2022). Randomization-based confidence intervals can be constructed by inverting FRTs (Luo et al., 2021; Zhu & Liu, 2023; Fiksel, 2024), and randomization inference can test bounded nulls and construct confidence intervals for treatment effect quantiles (Caughey et al., 2023). Extending these methods to hybrid controlled trials would be valuable.

## Software and Data

A user-friendly R package, `intFRT`, is available at: https://github.com/ke-zhu/intFRT.

## Acknowledgment

We thank the anonymous reviewers and meta-reviewers of ICML 2025 for their helpful comments, which significantly

improved the manuscript. This project is supported by the Food and Drug Administration (FDA) of the U.S. Department of Health and Human Services (HHS) as part of a financial assistance award U01FD007934 totaling $1,674,013 over two years funded by FDA/HHS. It is also supported by the National Institute On Aging of the National Institutes of Health under Award Number R01AG06688, totaling $1,565,763 over four years. The contents are those of the authors and do not necessarily represent the official views of, nor an endorsement by, FDA/HHS, the National Institutes of Health, or the U.S. Government.

## Impact Statement

This paper presents work aimed at advancing data integration, conformal inference, and their applications in biomedical science. The potential societal impact of this research is substantial, including fostering the reliable and efficient use of real-world data, accelerating drug development processes, improving the understanding of rare diseases, and ultimately enhancing patient outcomes.

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

# A. Additional conformal $p$-values

**Full conformal $p$-value.** Full conformal inference (Vovk et al., 2005) fully utilizes all data in $\mathcal{C}$ for both training and calibration. We can still use the absolute residual as the score function: $s_i = |Y_i - \hat{f}_j(X_i)|$ for $i \in \mathcal{C}$ and $s_j = |Y_j - \hat{f}_j(X_j)|$, where $\hat{f}_j(x)$ is a prediction model fitted by the augmented set $\mathcal{C} \cup \{j\}$. To measure the extremeness of observing $s_j$ under the exchangeability, we define the full conformal $p$-value as the proportion of the elements in $\{s_i\}_{i \in \mathcal{C}}$ that are larger than or equal to $s_j$, that is, $p_j^{\text{full}} = \{\sum_{i \in \mathcal{C}} \mathbb{I}(s_i \geq s_j) + 1\}/(|\mathcal{C}| + 1)$.

**Proposition A.1.** *For $j \in \mathcal{E}$, suppose that $(X_j, Y_j)$ and $\{(X_i, Y_i)\}_{i \in \mathcal{C}}$ are exchangeable. For $\gamma \in (0, 1)$, we have $\mathbb{P}(p_j^{\text{full}} \leq \gamma) \leq \gamma$. If $s_j$ and $\{s_i\}_{i \in \mathcal{C}}$ have distinct values, we have $\mathbb{P}(p_j^{\text{full}} \leq \gamma) = \left\{ \lfloor \gamma (|\mathcal{C}| + 1) \rfloor \right\} / (|\mathcal{C}| + 1) > \gamma - 1/(|\mathcal{C}| + 1)$.*

To compute full conformal $p$-values for all ECs $j \in \mathcal{E}$, the prediction model must be refit $n_\mathcal{E}$ times, which is time-consuming for large EC samples.

**Jackknife+ $p$-value.** Jackknife+ $p$-values (Barber et al., 2021) is a special case of CV+ with $K = |\mathcal{C}|$. We use the leave-one-out training set $\mathcal{C} \setminus \{i\}$ to fit prediction models $\hat{f}_{-i}(x)$ and use the absolute residual as the score function: $s_i = |Y_i - \hat{f}_{-i}(X_i)|$ and $s_j^{(i)} = |Y_j - \hat{f}_{-i}(X_j)|$ for $i \in \mathcal{C}$. We define the Jackknife+ $p$-value as the proportion of $\{s_i\}_{i \in \mathcal{C}}$ that are larger than the corresponding $\{s_j^{(i)}\}_{i \in \mathcal{C}}$, that is, $p_j^{\text{jackknife+}} = \{\sum_{i \in \mathcal{C}} \mathbb{I}(s_i \geq s_j^{(i)}) + 1\}/(|\mathcal{C}| + 1)$.

**Proposition A.2.** *For $j \in \mathcal{E}$, suppose that $(X_j, Y_j)$ and $\{(X_i, Y_i)\}_{i \in \mathcal{C}}$ are exchangeable. For $\gamma \in (0, 1)$, we have $\mathbb{P}(p_j^{\text{jackknife+}} \leq \gamma) \leq 2\gamma - 1/(|\mathcal{C}| + 1) < 2\gamma$.*

*Remark* A.3. The factor of 2 cannot be reduced without further assumptions, as shown by pathological cases in Barber et al. (2021), though the empirical error rate is close to $\gamma$.

# B. Proofs

## B.1. Proof of Theorem 2.3

*Proof of Theorem 2.3.* Under $H_0$, the imputed potential outcomes are the same as the true potential outcomes. Thus, the distribution of $T^* \equiv T(\boldsymbol{A}^*)$ is the same as that of $T \equiv T(\boldsymbol{A})$. With simplified notations, we have

$$\mathbb{P}_{\boldsymbol{A}}(p^{\text{FRT}} \leq \alpha) = \mathbb{P}_{\boldsymbol{A}} \left\{ \mathbb{P}_{\boldsymbol{A}^*} (T^* \geq T) \leq \alpha \right\}.$$

In a finite sample, $\boldsymbol{A}$ can take only a finite set of values, which implies that $T$ must also take on a finite set of values. Suppose these values are

$$T_1 > \ldots > T_m > \ldots > T_M,$$

and

$$\mathbb{P}_{\boldsymbol{A}}(T = T_m) = \mathbb{P}_{\boldsymbol{A}^*}(T^* = T_m) = \alpha_m, \quad m = 1, \ldots, M.$$

For $T \in \{T_1, \ldots, T_M\}$, we have $\alpha_1 \leq \mathbb{P}_{\boldsymbol{A}^*}(T^* \geq T) \leq \sum_{m=1}^M \alpha_m = 1$. If $0 < \alpha < \alpha_1$, we have

$$\mathbb{P}_{\boldsymbol{A}}(p^{\text{FRT}} \leq \alpha) = \mathbb{P}_{\boldsymbol{A}} \left\{ \mathbb{P}_{\boldsymbol{A}^*} (T^* \geq T) \leq \alpha \right\} = 0 \leq \alpha.$$

If $\alpha_1 \leq \alpha < 1$, $\exists \tilde{M} \in \{1, \ldots, M-1\}$, such that $\sum_{m=1}^{\tilde{M}} \alpha_m \leq \alpha$ and $\sum_{m=1}^{\tilde{M}+1} \alpha_m > \alpha$. Then, we have

$$\mathbb{P}_{\boldsymbol{A}}(p^{\text{FRT}} \leq \alpha) = \mathbb{P}_{\boldsymbol{A}} \left\{ \mathbb{P}_{\boldsymbol{A}^*} (T^* \geq T) \leq \alpha \right\} = \mathbb{P}_{\boldsymbol{A}} \left\{ T \in \{T_1, \ldots, T_{\tilde{M}}\} \right\} = \sum_{m=1}^{\tilde{M}} \alpha_m \leq \alpha.$$

If $T(\boldsymbol{A})$ takes distinct values for different $\boldsymbol{A} \in \mathcal{A}$, $p^{\text{FRT}}$ is uniformly distributed:

$$\mathbb{P}_{\boldsymbol{A}} \left( p^{\text{FRT}} = \frac{a}{|\mathcal{A}|} \right) = \frac{1}{|\mathcal{A}|}, \quad a = 1, \ldots, |\mathcal{A}|.$$

Thus, we have

$$\mathbb{P}_{\boldsymbol{A}}(p^{\text{FRT}} \leq \alpha) = \frac{\lfloor \alpha |\mathcal{A}| \rfloor}{|\mathcal{A}|} > \frac{\alpha |\mathcal{A}| - 1}{|\mathcal{A}|} = \alpha - \frac{1}{|\mathcal{A}|}.$$

$\square$

*Remark* B.1. If $T$ is a continuous random variable, suppose its distribution function is $F(t) = P(T \leq t)$, then the proof could be simplified as

$$
\begin{aligned}
\mathbb{P}_{\boldsymbol{A}}\left\{\mathbb{P}_{\boldsymbol{A}^*}\left(T^* \geq T\right) \leq \alpha\right\} &= P\left\{1 - F(T) \leq \alpha\right\} \\
&= P\left\{T \geq F^{-1}(1 - \alpha)\right\} \\
&= 1 - F\{F^{-1}(1 - \alpha)\} \\
&= \alpha.
\end{aligned}
$$

However, $T$ is discrete with finite values, and we provide a rigorous proof in the finite-sample setting.

## B.2. Proof of Theorem 2.4

We invoke two lemmas from the Supplementary Material of Puelz et al. (2022).

**Lemma B.2** (Lemma 5 in Puelz et al. (2022)). *Suppose Assumptions (b) and (c) in Theorem 2.4 hold, for some $r \in (0.5, 1 + O(\log^{-1} M))$, we have*
$$
\mathbb{E}(F_{1,n}(q_\alpha) - F_{1,n}(q_{\alpha,M})) \geq -O(M^{-r}),
$$

**Lemma B.3** (Lemma 4 in Puelz et al. (2022)). *Suppose Assumption (a) of Theorem 2.4 holds, for any $0 < \iota < 0.5$ and large enough $M$, we have*
$$
\mathbb{E}(F_{1,n,M}(z) - F_{1,n}(z)) = O(M^{-0.5+\iota}), \quad \text{for any } z \in \mathbb{R}.
$$

*Proof of Theorem 2.4.* Let $q_{\alpha,M} = F_{0,n,M}^{-1}(1 - \alpha)$ and $q_\alpha = F_{0,n}^{-1}(1 - \alpha)$. Thus, we have

$$
\begin{aligned}
\psi_{N,M} &= 1 - F_{1,n,M}\left(F_{0,n,M}^{-1}(1 - \alpha)\right) \\
&= 1 - F_{1,n,M}\left(q_{\alpha,M}\right) \\
&= \underbrace{1 - F_{1,n}(q_\alpha)}_{T_1} + \underbrace{F_{1,n}(q_\alpha) - F_{1,n}(q_{\alpha,M})}_{T_2} + \underbrace{F_{1,n}(q_{\alpha,M}) - F_{1,n,M}(q_{\alpha,M})}_{T_3}.
\end{aligned}
\tag{3}
$$

By Assumptions (b) and (c), we have

$$
T_1 = 1 - F_{1,n}\left(F_{0,n}^{-1}(1 - \alpha)\right) = 1 - F\left(F^{-1}(1 - \alpha) - \tau/\sigma_N\right).
$$

Combined with Lemmas B.2 and B.3, we have

$$
\mathbb{E}(\psi_{N,M}) \geq 1 - F\left(F^{-1}(1 - \alpha) - \tau/\sigma_N\right) - O(M^{-r}) - O(M^{-0.5+\iota}).
$$

The result follows from that $r > 0.5 > 0.5 - \iota > 0$. □

## B.3. Proof of Proposition A.1

*Proof of Proposition A.1.* Since the calibration set $(X_i, Y_j)_{i \in \mathcal{C}}$ and external control $(X_j, Y_j)$ are exchangeable, we have $(s_i)_{i \in \mathcal{C}}$ and $s_j$ are exchangeable. Thus, we have

$$
\begin{aligned}
\mathbb{P}(p_j^{\text{full}} \leq \gamma) &= \mathbb{P}\left(\frac{\sum_{i \in \mathcal{C}} \mathbb{I}(s_i \geq s_j) + 1}{|\mathcal{C}| + 1} \leq \gamma\right) \\
&\leq \frac{\lfloor \gamma(|\mathcal{C}| + 1)\rfloor}{|\mathcal{C}| + 1} \\
&\leq \gamma,
\end{aligned}
$$

where the first inequality is due to exchangeability and the possibility of ties in $(s_i)_{i \in \mathcal{C}}$ and $s_j$.

If $s_j$ and $\{s_i\}_{i \in \mathcal{C}}$ have distinct values, $p_j^{\text{full}}$ is uniformly distributed due to exchangeability. That is,

$$
\mathbb{P}\left(p_j^{\text{full}} = \frac{a}{|\mathcal{C}| + 1}\right) = \frac{1}{|\mathcal{C}| + 1}, \quad a = 1, \ldots, |\mathcal{C}| + 1.
$$

Thus, we have

$$
\mathbb{P}(p_j^{\text{full}} \leq \gamma) = \frac{\lfloor \gamma(|\mathcal{C}| + 1)\rfloor}{|\mathcal{C}| + 1} > \frac{\gamma(|\mathcal{C}| + 1) - 1}{|\mathcal{C}| + 1} = \gamma - \frac{1}{|\mathcal{C}| + 1}.
$$

□

### B.4. Proof of Proposition 3.1

*Proof of Proposition 3.1.* Since the calibration set $(X_i, Y_j)_{i \in \mathcal{C}}$ and external control $(X_j, Y_j)$ are exchangeable, we have $(s_i)_{i \in \mathcal{C}_1}$ and $s_j$ are exchangeable. Thus, we have

$$
\begin{aligned}
\mathbb{P}(p_j^{\text{split}} \leq \gamma) &= \mathbb{P}\left( \frac{\sum_{i \in \mathcal{C}_1} \mathbb{I}(s_i \geq s_j) + 1}{|\mathcal{C}_1| + 1} \leq \gamma \right) \\
&\leq \frac{\lfloor \gamma(|\mathcal{C}_1| + 1) \rfloor}{|\mathcal{C}_1| + 1} \\
&\leq \gamma,
\end{aligned}
$$

where the first inequality is due to exchangeability and the possibility of ties in $(s_i)_{i \in \mathcal{C}_1}$ and $s_j$.

If $s_j$ and $\{s_i\}_{i \in \mathcal{C}_1}$ have distinct values, $p_j^{\text{split}}$ is uniformly distributed due to exchangeability. That is,

$$
\mathbb{P}\left( p_j^{\text{split}} = \frac{a}{|\mathcal{C}_1| + 1} \right) = \frac{1}{|\mathcal{C}_1| + 1}, \quad a = 1, \ldots, |\mathcal{C}_1| + 1.
$$

Thus, we have

$$
\mathbb{P}(p_j^{\text{split}} \leq \gamma) = \frac{\lfloor \gamma(|\mathcal{C}_1| + 1) \rfloor}{|\mathcal{C}_1| + 1} > \frac{\gamma(|\mathcal{C}_1| + 1) - 1}{|\mathcal{C}_1| + 1} = \gamma - \frac{1}{|\mathcal{C}_1| + 1}.
$$

$\square$

### B.5. Proof of Proposition A.2

**Lemma B.4.** *Consider a matrix $R \in \mathbb{R}^{(n+1) \times (n+1)}$ with elements $R_{ij}$. Define the set*

$$
\mathcal{S} = \left\{ j \in \{1, \ldots, n+1\} : \sum_{i=1}^{n+1} \mathbb{I}(R_{ij} < R_{ji}) \geq (1 - \gamma)(n+1) \right\}, \quad \gamma \in (0, 1).
$$

*Then, we have*

$$
s \leq 2\gamma(n+1) - 1 < 2\gamma(n+1),
$$

*where $s = |\mathcal{S}|$.*

*Proof.* Since

$$
\sum_{i=1}^{n+1} \mathbb{I}(R_{ij} < R_{ji}) \geq (1 - \gamma)(n+1) \quad \Leftrightarrow \quad \sum_{i=1}^{n+1} \mathbb{I}(R_{ij} \geq R_{ji}) \leq \gamma(n+1),
$$

by summing over all $j \in \mathcal{S}$, we have

$$
\sum_{j \in \mathcal{S}} \sum_{i=1}^{n+1} \mathbb{I}(R_{ij} \geq R_{ji}) \leq s\gamma(n+1).
$$

For $i \neq j$, since $\mathbb{I}(R_{ij} \geq R_{ji}) + \mathbb{I}(R_{ji} \geq R_{ij}) \geq 1$, we have

$$
\begin{aligned}
\sum_{j \in \mathcal{S}} \sum_{i \in \mathcal{S}} \mathbb{I}(R_{ij} \geq R_{ji}) &= \sum_{j \in \mathcal{S}} \sum_{i \in \mathcal{S}, i \neq j} \mathbb{I}(R_{ij} \geq R_{ji}) + s \\
&\geq \frac{s(s-1)}{2} + s.
\end{aligned}
$$

By combining these two inequalities, we obtain

$$
\begin{aligned}
\frac{s(s-1)}{2} + s &\leq \sum_{j \in \mathcal{S}} \sum_{i \in \mathcal{S}} \mathbb{I}(R_{ij} \geq R_{ji}) \\
&\leq \sum_{j \in \mathcal{S}} \sum_{i=1}^{n+1} \mathbb{I}(R_{ij} \geq R_{ji}) \\
&\leq s\gamma(n+1).
\end{aligned}
$$

Thus, we have

$$\frac{(s-1)}{2} + 1 \le \gamma(n+1) \quad \Rightarrow \quad s \le 2\gamma(n+1) - 1 < 2\gamma(n+1).$$

$\square$

*Proof of Proposition A.2.* For $i', j' \in \mathcal{C} \cup \{j\}$, we define

$$R_{i'j'} = \begin{cases} +\infty & i' = j', \\ \left| Y_{i'} - \hat{f}_{-(i',j')}(X_{i'}) \right| & i' \ne j', \end{cases}$$

where $\hat{f}_{-(i',j')}$ is a prediction model fitted by the leave-two-out augmented set $(\mathcal{C} \cup \{j\}) \setminus \{i', j'\}$. For $i \in \mathcal{C}$, since $(\mathcal{C} \cup \{j\}) \setminus \{i, j\} = \mathcal{C} \setminus \{i\}$, we have $\hat{f}_{-i}(x) = \hat{f}_{-(i,j)}(x)$, thereby,

$$s_i = |Y_i - \hat{f}_{-i}(X_i)| = R_{ij},$$
$$s_j^{(i)} = |Y_j - \hat{f}_{-i}(X_j)| = R_{ji}.$$

Thus, we have

$$\begin{aligned} \mathbb{P}(p_j^{\text{jackknife+}} \le \gamma) &= \mathbb{P}\left( \frac{\sum_{i\in\mathcal{C}} \mathbb{I}(s_i \ge s_j^{(i)}) + 1}{|\mathcal{C}| + 1} \le \gamma \right) \\ &= \mathbb{P}\left( \frac{\sum_{i\in\mathcal{C}\cup\{j\}} \mathbb{I}(R_{ij} \ge R_{ji})}{|\mathcal{C}| + 1} \le \gamma \right) \\ &= \mathbb{P}\left( \sum_{i\in\mathcal{C}\cup\{j\}} \mathbb{I}(R_{ij} < R_{ji}) \ge (1-\gamma)(|\mathcal{C}| + 1) \right) \\ &\le 2\gamma - \frac{1}{|\mathcal{C}| + 1} \\ &< 2\gamma, \end{aligned}$$

where first inequality is due to exchangeability and Lemma B.4. $\square$

## B.6. Proof of Proposition 3.3

**Lemma B.5.** *Suppose $m = n/K$ is an integer, and the $n + m$ units are evenly divided into $K + 1$ sets, denoted by $\mathcal{C}_1, \ldots, \mathcal{C}_{K+1}$. Consider a matrix $R \in \mathbb{R}^{(n+m)\times(n+m)}$ with elements $R_{ij} = R_{ji}$ if $i$ and $j$ belong to the same set. Define the set*

$$\mathcal{S} = \left\{ j \in \{1, \ldots, n+m\} : \sum_{i=1}^{n+m} \mathbb{I}(R_{ij} < R_{ji}) \ge (1-\gamma)(n+1) \right\}, \quad \gamma \in (0, 1).$$

*Then, we have*

$$s \le 2\gamma(n+1) + m - 2,$$

*where $s = |\mathcal{S}|$.*

*Proof.* For $j \in \mathcal{S}$, by definition, we have

$$\sum_{i=1}^{n+m} \mathbb{I}(R_{ij} \ge R_{ji}) \le (n+m) - (1-\gamma)(n+1).$$

Since $R_{ij} = R_{ji}$ if $i$ and $j$ belong to the same set, we have

$$\begin{aligned} \sum_{i=1}^{n+m} \mathbb{I}(R_{ij} \ge R_{ji}) &= \sum_{i\notin\mathcal{C}_{k(j)}} \mathbb{I}(R_{ij} \ge R_{ji}) + \sum_{i\in\mathcal{C}_{k(j)}} \mathbb{I}(R_{ij} \ge R_{ji}) \\ &= \sum_{i\notin\mathcal{C}_{k(j)}} \mathbb{I}(R_{ij} \ge R_{ji}) + m, \end{aligned}$$

where $\mathcal{C}_{k(j)}$ is the set containing unit $j$. Thus, we have

$$\sum_{i \notin \mathcal{C}_{k(j)}} \mathbb{I}(R_{ij} \geq R_{ji}) \leq (n+m) - (1-\gamma)(n+1) - m$$

$$= \gamma(n+1) - 1.$$

By summing over all $j \in \mathcal{S}$, we have

$$\sum_{j \in \mathcal{S}} \sum_{i \notin \mathcal{C}_{k(j)}} \mathbb{I}(R_{ij} \geq R_{ji}) \leq s\{\gamma(n+1) - 1\}. \tag{4}$$

On the other hand, for $i \neq j$, since $\mathbb{I}(R_{ij} \geq R_{ji}) + \mathbb{I}(R_{ji} \geq R_{ij}) \geq 1$, we have

$$\sum_{j \in \mathcal{S}} \sum_{i \in \mathcal{S}, i \neq j} \mathbb{I}(R_{ij} \geq R_{ji}) \geq \frac{s(s-1)}{2}.$$

Since $R_{ij} = R_{ji}$ if $i$ and $j$ belong to the same set, we have

$$\sum_{j \in \mathcal{S}} \sum_{i \in \mathcal{S}, i \neq j} \mathbb{I}(R_{ij} \geq R_{ji}) = \sum_{j \in \mathcal{S}} \sum_{i \in \mathcal{S}, i \notin \mathcal{C}_{k(j)}} \mathbb{I}(R_{ij} \geq R_{ji}) + \sum_{j \in \mathcal{S}} \sum_{i \in \mathcal{S}, i \in \mathcal{C}_{k(j)}, i \neq j} \mathbb{I}(R_{ij} \geq R_{ji})$$

$$= \sum_{j \in \mathcal{S}} \sum_{i \in \mathcal{S}, i \notin \mathcal{C}_{k(j)}} \mathbb{I}(R_{ij} \geq R_{ji}) + \sum_{k=1}^{K+1} \frac{s_k(s_k - 1)}{2},$$

where $s_k = |\mathcal{C}_k \cap \mathcal{S}|$. Thus, we have

$$\sum_{j \in \mathcal{S}} \sum_{i \in \mathcal{S}, i \notin \mathcal{C}_{k(j)}} \mathbb{I}(R_{ij} \geq R_{ji}) \geq \frac{s(s-1)}{2} - \sum_{k=1}^{K+1} \frac{s_k(s_k - 1)}{2}. \tag{5}$$

By combining (4) and (5), we have

$$\frac{s(s-1)}{2} - \sum_{k=1}^{K+1} \frac{s_k(s_k - 1)}{2} \leq \sum_{j \in \mathcal{S}} \sum_{i \in \mathcal{S}, i \notin \mathcal{C}_{k(j)}} \mathbb{I}(R_{ij} \geq R_{ji})$$

$$\leq \sum_{j \in \mathcal{S}} \sum_{i \notin \mathcal{C}_{k(j)}} \mathbb{I}(R_{ij} \geq R_{ji})$$

$$\leq s\{\gamma(n+1) - 1\}.$$

Since $s_k \leq m$, we have

$$\sum_{k=1}^{K+1} \frac{s_k(s_k - 1)}{2} \leq \frac{s(m-1)}{2}.$$

Thus, we have

$$s \leq 2\gamma(n+1) + m - 2.$$

$\square$

*Proof of Proposition 3.3.* We consider $m = |\mathcal{C}|/K$ is an integer for simplicity. Let $\mathcal{C}_{K+1}$ contain $j$ and other $m-1$ hypothetical points. For $i', j' \in \cup_{k=1}^{K+1} \mathcal{C}_k$, we define

$$R_{i'j'} = \begin{cases} +\infty & k(i') = k(j'), \\ \left| Y_{i'} - \hat{f}_{-(\mathcal{C}_{k(i')}, \mathcal{C}_{k(j')})}(X_{i'}) \right| & k(i') \neq k(j'), \end{cases}$$

where $\hat{f}_{-(\mathcal{C}_{k(i')},\mathcal{C}_{k(j')})}$ is a prediction model fitted by the leave-two-set-out augmented set $(\cup_{k=1}^{K+1}\mathcal{C}_k) \setminus (\mathcal{C}_{k(i')} \cup \mathcal{C}_{k(j')})$. Since $\mathcal{C} = \cup_{k=1}^{K}\mathcal{C}_k$ and $\mathcal{C}_{k(j)} = \mathcal{C}_{K+1}$, we have $(\cup_{k=1}^{K+1}\mathcal{C}_k) \setminus (\mathcal{C}_{k(i)} \cup \mathcal{C}_{k(j)}) = \mathcal{C} \setminus \mathcal{C}_{k(i)}$ for $i \in \mathcal{C}$. Thus, for $i \in \mathcal{C}$, we have $\hat{f}_{-\mathcal{C}_{k(i)}}(x) = \hat{f}_{-(\mathcal{C}_{k(i)},\mathcal{C}_{k(j)})}(x)$, thereby,

$$s_i = |Y_i - \hat{f}_{-\mathcal{C}_{k(i)}}(X_i)| = R_{ij},$$
$$s_j^{(i)} = |Y_j - \hat{f}_{-\mathcal{C}_{k(i)}}(X_j)| = R_{ji}.$$

Thus, we have

$$
\begin{aligned}
\mathbb{P}(p_j^{\text{cv}+} \leq \gamma) &= \mathbb{P}\left( \frac{\sum_{i\in\mathcal{C}} \mathbb{I}(s_i \geq s_j^{(i)}) + 1}{|\mathcal{C}| + 1} \leq \gamma \right) \\
&= \mathbb{P}\left( \frac{\sum_{i\in\mathcal{C}\cup\{j\}} \mathbb{I}(R_{ij} \geq R_{ji})}{|\mathcal{C}| + 1} \leq \gamma \right) \\
&= \mathbb{P}\left( \sum_{i\in\mathcal{C}\cup\{j\}} \mathbb{I}(R_{ij} < R_{ji}) \geq (1-\gamma)(|\mathcal{C}| + 1) \right) \\
&\leq \mathbb{P}\left( \sum_{i\in\cup_{k=1}^{K+1}\mathcal{C}_k} \mathbb{I}(R_{ij} < R_{ji}) \geq (1-\gamma)(|\mathcal{C}| + 1) \right) \\
&\leq \frac{2\gamma(|\mathcal{C}| + 1) + m - 2}{|\mathcal{C}| + m} \\
&\leq 2\gamma + \frac{(1-2\gamma)(m-1) - 1}{|\mathcal{C}| + m} \\
&\leq 2\gamma + \frac{1 - K/|\mathcal{C}|}{K + 1},
\end{aligned}
$$

where the second inequality is due to exchangeability and Lemma B.5. $\qquad\square$

## B.7. Proof of Theorem 3.4

*Proof of Theorem 3.4.* Since $\epsilon_\gamma$ is a centered sub-Gaussian variable with parameter $\phi_\gamma$, we have $\epsilon_\gamma - \epsilon_1$ as a centered sub-Gaussian variable with parameter $2\Phi$, where $\Phi = \max_{\gamma\in\Gamma} \phi_\gamma$. Moreover, we have $(\epsilon_\gamma - \epsilon_1)^2 - \kappa_\gamma^2$ is a centered sub-exponential variable with parameters $(c_1\Phi^2, c_1\Phi^2)$, where $c_1$ is a constant. By $\hat{\tau}_\gamma - \hat{\tau}_1 = (\delta_\gamma - \delta_1) + (\epsilon_\gamma - \epsilon_1)$ and using the concentration inequalities for sub-Gaussian and sub-exponential variables (Wainwright, 2019), it follows that, with probability at least $1 - 4\iota$,

$$
\begin{aligned}
&\max_{\gamma\in\Gamma} |(\hat{\tau}_\gamma - \hat{\tau}_1)^2 - (\delta_\gamma - \delta_1)^2 - \kappa_\gamma^2| \\
&= \max_{\gamma\in\Gamma} |2(\delta_\gamma - \delta_1)(\epsilon_\gamma - \epsilon_1) + (\epsilon_\gamma - \epsilon_1)^2 - \kappa_\gamma^2| \\
&\leq 8\sqrt{2}\Delta\Phi\sqrt{\log(|\Gamma|/\iota)} + \max\left\{ \sqrt{2}c_1\Phi^2\sqrt{\log(|\Gamma|/\iota)},\ 2c_1\Phi^2\log(|\Gamma|/\iota) \right\},
\end{aligned}
\tag{6}
$$

where $\Delta = \max_{\gamma\in\Gamma} |\delta_\gamma|$.

By (6), it follows that, with probability at least $1 - 4\iota$,

$$
\max_{\gamma \in \Gamma} \left| \widehat{\mathrm{MSE}}(\gamma) - \mathrm{MSE}(\gamma) \right|
$$

$$
= \max_{\gamma \in \Gamma} \left| (\hat{\tau}_\gamma - \hat{\tau}_1)^2 - \widehat{\mathbb{V}}(\hat{\tau}_\gamma - \hat{\tau}_1) + \widehat{\mathbb{V}}(\hat{\tau}_\gamma) - \delta_\gamma^2 - \sigma_\gamma^2 \right|
$$

$$
\leq \max_{\gamma \in \Gamma} \left| (\hat{\tau}_\gamma - \hat{\tau}_1)^2 - \kappa_\gamma^2 + \sigma_\gamma^2 - \delta_\gamma^2 - \sigma_\gamma^2 \right| + \max_{\gamma \in \Gamma} |\widehat{\mathbb{V}}(\hat{\tau}_\gamma - \hat{\tau}_1) - \kappa_\gamma^2| + \max_{\gamma \in \Gamma} |\widehat{\mathbb{V}}(\hat{\tau}_\gamma) - \sigma_\gamma^2|
$$

$$
\leq \max_{\gamma \in \Gamma} \left| (\delta_\gamma - \delta_1)^2 - \delta_\gamma^2 \right| + c\Delta\Phi\sqrt{\log\left(|\Gamma|/\iota\right)} + \max\left\{ c\Phi^2\sqrt{\log\left(|\Gamma|/\iota\right)},\ c\Phi^2\log\left(|\Gamma|/\iota\right) \right\}
$$

$$
+ \max_{\gamma \in \Gamma} |\widehat{\mathbb{V}}(\hat{\tau}_\gamma - \hat{\tau}_1) - \kappa_\gamma^2| + \max_{\gamma \in \Gamma} |\widehat{\mathbb{V}}(\hat{\tau}_\gamma) - \sigma_\gamma^2|
$$

$$
\leq c\Delta|\delta_1| + c\Delta\Phi\sqrt{\log\left(|\Gamma|/\iota\right)} + \max\left\{ c\Phi^2\sqrt{\log\left(|\Gamma|/\iota\right)},\ c\Phi^2\log\left(|\Gamma|/\iota\right) \right\}
$$

$$
+ \max_{\gamma \in \Gamma} |\widehat{\mathbb{V}}(\hat{\tau}_\gamma - \hat{\tau}_1) - \kappa_\gamma^2| + \max_{\gamma \in \Gamma} |\widehat{\mathbb{V}}(\hat{\tau}_\gamma) - \sigma_\gamma^2|,
$$

where $c$ is a constant. $\qquad\square$

### B.8. Proof of Theorem 3.5

*Proof of Theorem 3.5.* Since $\epsilon_\gamma$ is a centered sub-Gaussian variable with parameter $\phi_\gamma$, we have $\epsilon_\gamma^2 - \sigma_\gamma^2$ as a centered sub-exponential variable with parameter $(c_2\Phi^2, c_2\Phi^2)$, where $c_2$ is a constant. By $\hat{\tau}_\gamma - \tau = \delta_\gamma + \epsilon_\gamma$ and using the concentration inequalities for sub-Gaussian and sub-exponential variables (Wainwright, 2019), it follows that, with probability at least $1 - 4\iota$,

$$
\max_{\gamma \in \Gamma} |(\hat{\tau}_\gamma - \tau)^2 - \delta_\gamma^2 - \sigma_\gamma^2|
$$

$$
= \max_{\gamma \in \Gamma} |2\delta_\gamma\epsilon_\gamma + \epsilon_\gamma^2 - \sigma_\gamma^2|
$$

$$
\leq 2\sqrt{2}\Delta\Phi\sqrt{\log\left(|\Gamma|/\iota\right)} + \max\left\{ \sqrt{2}c_2\Phi^2\sqrt{\log\left(|\Gamma|/\iota\right)},\ 2c_2\Phi^2\log\left(|\Gamma|/\iota\right) \right\}. \tag{7}
$$

By (6) and (7), it follows that, with probability at least $1 - 8\iota$,

$$
\max_{\gamma \in \Gamma} \left| \widehat{\mathrm{MSE}}(\gamma) - (\hat{\tau}_\gamma - \tau)^2 \right|
$$

$$
= \max_{\gamma \in \Gamma} \left| (\hat{\tau}_\gamma - \hat{\tau}_1)^2 - \widehat{\mathbb{V}}(\hat{\tau}_\gamma - \hat{\tau}_1) + \widehat{\mathbb{V}}(\hat{\tau}_\gamma) - (\hat{\tau}_\gamma - \tau)^2 \right|
$$

$$
\leq \max_{\gamma \in \Gamma} \left| (\hat{\tau}_\gamma - \hat{\tau}_1)^2 - \kappa_\gamma^2 + \sigma_\gamma^2 - (\hat{\tau}_\gamma - \tau)^2 \right| + \max_{\gamma \in \Gamma} |\widehat{\mathbb{V}}(\hat{\tau}_\gamma - \hat{\tau}_1) - \kappa_\gamma^2| + \max_{\gamma \in \Gamma} |\widehat{\mathbb{V}}(\hat{\tau}_\gamma) - \sigma_\gamma^2|
$$

$$
\leq \max_{\gamma \in \Gamma} \left| (\delta_\gamma - \delta_1)^2 - \delta_\gamma^2 \right| + c\Delta\Phi\sqrt{\log\left(|\Gamma|/\iota\right)} + \max\left\{ c\Phi^2\sqrt{\log\left(|\Gamma|/\iota\right)},\ c\Phi^2\log\left(|\Gamma|/\iota\right) \right\}
$$

$$
+ \max_{\gamma \in \Gamma} |\widehat{\mathbb{V}}(\hat{\tau}_\gamma - \hat{\tau}_1) - \kappa_\gamma^2| + \max_{\gamma \in \Gamma} |\widehat{\mathbb{V}}(\hat{\tau}_\gamma) - \sigma_\gamma^2|
$$

$$
\leq c\Delta|\delta_1| + c\Delta\Phi\sqrt{\log\left(|\Gamma|/\iota\right)} + \max\left\{ c\Phi^2\sqrt{\log\left(|\Gamma|/\iota\right)},\ c\Phi^2\log\left(|\Gamma|/\iota\right) \right\}
$$

$$
+ \max_{\gamma \in \Gamma} |\widehat{\mathbb{V}}(\hat{\tau}_\gamma - \hat{\tau}_1) - \kappa_\gamma^2| + \max_{\gamma \in \Gamma} |\widehat{\mathbb{V}}(\hat{\tau}_\gamma) - \sigma_\gamma^2|, \tag{8}
$$

where $c$ is a constant.

Since

$$
\left| \min_{\gamma \in \Gamma} \widehat{\mathrm{MSE}}(\gamma) - \min_{\gamma \in \Gamma}(\hat{\tau}_\gamma - \tau)^2 \right| \leq \max_{\gamma \in \Gamma} \left| \widehat{\mathrm{MSE}}(\gamma) - (\hat{\tau}_\gamma - \tau)^2 \right|,
$$

and

$$
\left| \min_{\gamma \in \Gamma} \widehat{\mathrm{MSE}}(\gamma) - (\hat{\tau}_{\hat{\gamma}} - \tau)^2 \right| = \left| \widehat{\mathrm{MSE}}(\hat{\gamma}) - (\hat{\tau}_{\hat{\gamma}} - \tau)^2 \right| \leq \max_{\gamma \in \Gamma} \left| \widehat{\mathrm{MSE}}(\gamma) - (\hat{\tau}_\gamma - \tau)^2 \right|,
$$

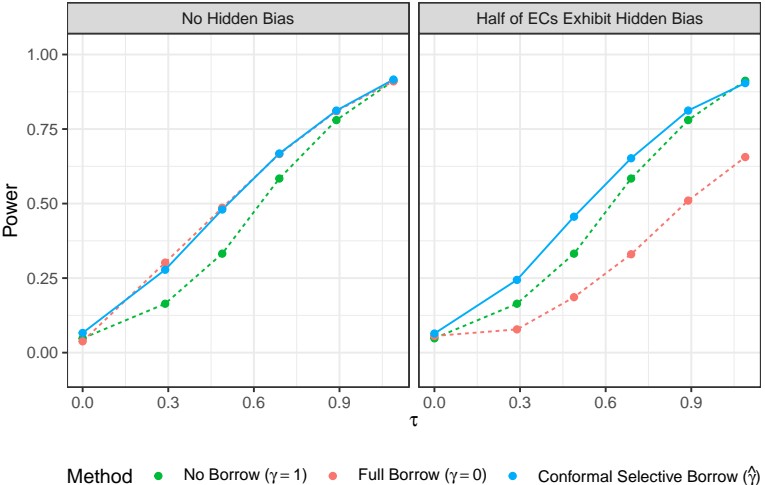

*Figure 3.* Power curves when $b = 0$ and $b = 8$.

we have

$$(\hat{\tau}_{\hat{\gamma}} - \tau)^2 - \min_{\gamma \in \Gamma}(\hat{\tau}_\gamma - \tau)^2 \leq 2 \max_{\gamma \in \Gamma} \left| \widehat{\text{MSE}}(\gamma) - (\hat{\tau}_\gamma - \tau)^2 \right|.$$

The result follows from (8). □

## C. Additional simulation results

### C.1. Power curve

For the scenario where there is no hidden bias ($b = 0$) and another where half of the ECs exhibit hidden bias with a magnitude of $b = 8$, we vary $\tau$ to plot the power curve, as shown in Figure 3. CSB outperforms NB in both cases, while FB demonstrates low power in the presence of hidden bias.

### C.2. Adaptivity of the selection threshold

Figure 4 illustrates how $\hat{\gamma}$ changes with the magnitude of $b$: (i) When there is no bias ($b = 0$), $\hat{\gamma}$ approaches 0 to borrow all ECs and maximize power; (ii) with moderate bias ($b = 1, 2, 3$), where distinguishing between biased and unbiased ECs is challenging, $\hat{\gamma}$ increases to help discard the biased ECs; (iii) when the bias is large ($b \geq 4$), $\hat{\gamma}$ decreases but remains non-zero, retaining more unbiased ECs, while easily discarding the biased ones.

### C.3. Various selection thresholds

Figure 5 shows the performance of the fixed selection threshold $\gamma$ and the adaptive selection threshold $\hat{\gamma}$ when $n_{\mathcal{E}} = 50$. As discussed in Section 3.3, smaller $\gamma$ selects more ECs but risks greater bias when distinguishing between biased and unbiased ECs is difficult. This creates a power trade-off across different bias levels, similar to MSE simulation results in data integration (Yang et al., 2023; Oberst et al., 2022; Lin et al., 2024). We find that (i) CSB with $\gamma = 0.6$ improves power compared to NB, except in extreme cases like $b = 2, 3$, where it decreases power slightly, and (ii) CSB with $\hat{\gamma}$ further improves power but also risks power loss in difficult scenarios. The power trade-off does not compromise the Type I error rate, which remains controlled with all selection thresholds.

### C.4. Comparison to Adaptive Lasso Selective Borrowing

Figure 6 presents the simulation results for ALSB with asymptotic inference. Unlike CSB + FRT, ALSB with asymptotic inference fails to control the type I error rate in this small sample size scenario. Additionally, CSB demonstrates better estimation and selection performance in most cases.

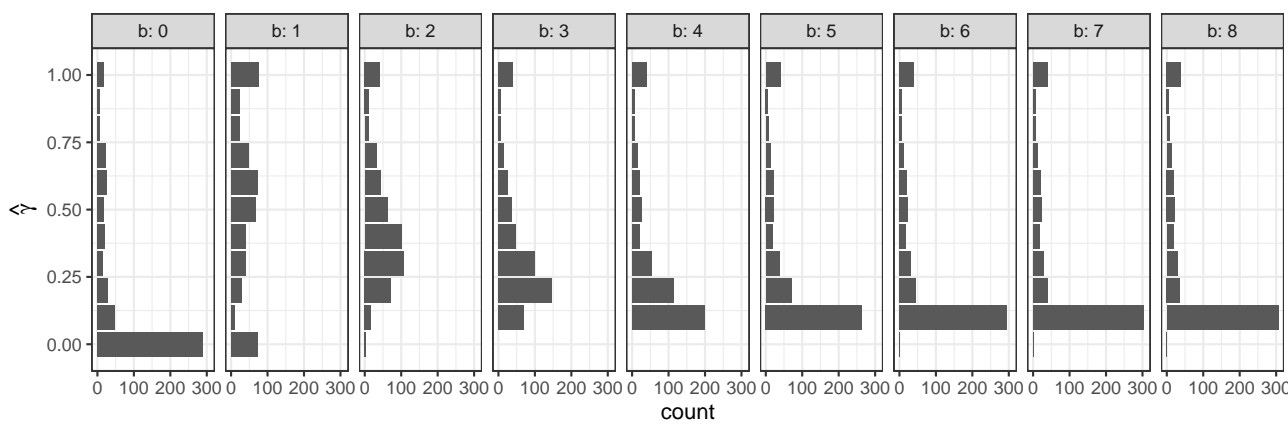

*Figure 4.* $\hat{\gamma}$ versus $b$ when $n_{\mathcal{E}} = 50$.

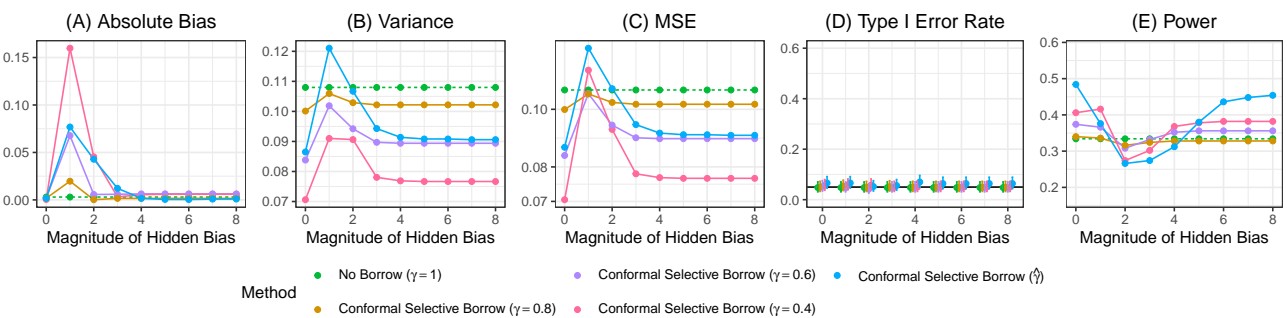

*Figure 5.* Simulation results for various selection threshold $\gamma$'s when $n_{\mathcal{E}} = 50$.

We further compared `CSB` with asymptotic inference to `ALSB` with asymptotic inference. Figure 7 shows that CSB+Asym Inf generally achieves better Type I error control than `ALSB+Asym Inf`, while performing comparably when $b = 1$.

We did not compare to `ALSB + FRT` because, while `CSB` is compatible with FRT, `ALSB` is not readily applicable due to its computational complexity. This highlights an advantage of `CSB` when exact finite-sample inference is desired.

### C.5. A larger sample size of ECs

Figures 8, 9, 10, 11, and 12 show the simulation results for $n_{\mathcal{E}} = 300$. The conclusion is similar to that in the main text.

### C.6. Dependent covariates with $p = 5$

We additionally consider $p = 5$ and $X \sim N(0, \Sigma)$, where $\Sigma$ is a Toeplitz matrix with $\rho = 0.6$ to introduce dependence among the coordinates of $X$. We did not consider larger $p$ since the sample size is small, with only 25 RCT controls. The simulation results (see Figure 13) show similar conclusions and demonstrate the robustness of our method.

## D. More details about the real data

**Pseudo-observations.** Figure 14 shows the pseudo-observations versus censored times for CALGB 9633 and NCDB, illustrating that (i) all pseudo-observations are less than or equal to the truncation time of 3 years; (ii) when an event occurs before 3 years, pseudo-observations are generally equal to the event time; and (iii) when censoring occurs before 3 years, pseudo-observations are typically greater than the censored time.

**Matching.** We use nearest-neighbor matching to mitigate the covariate imbalance between CALGB 9633 and NCDB. Tumor size was imputed for eight missing values in CALGB 9633 using the median of 4. NCDB samples with missing values

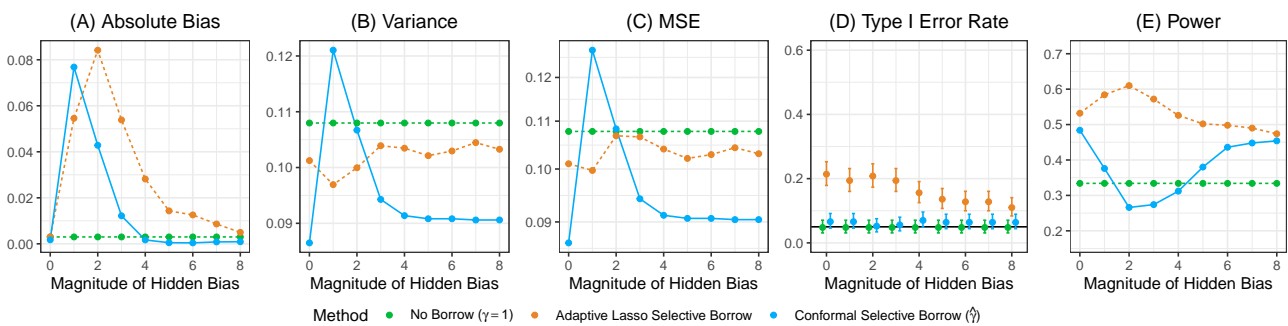

*Figure 6.* Comparison of CSB + FRT and ALSB + asymptotic inference when $n_{\mathcal{E}} = 50$.

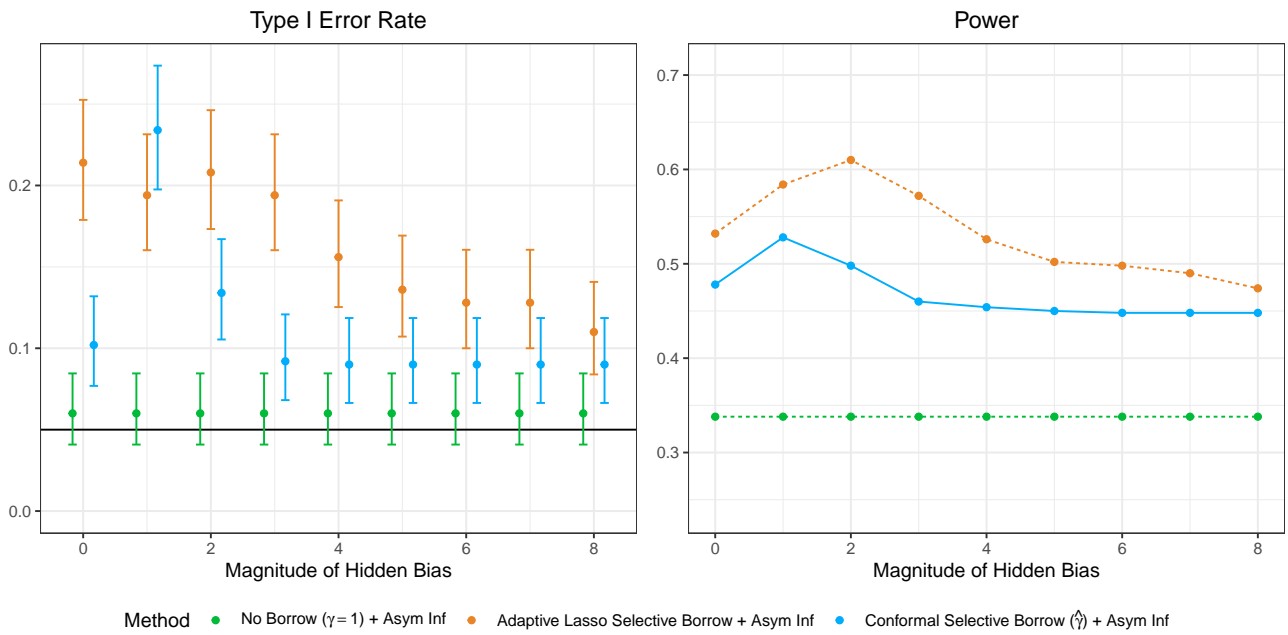

*Figure 7.* Comparison of CSB + asymptotic inference and ALSB + asymptotic inference when $n_{\mathcal{E}} = 50$.

or covariates outside the CALGB 9633 range were excluded, leaving 10,241 samples. We perform 1:1 nearest-neighbor matching using `MatchIt` (Ho et al., 2011), treating the sampling indicator $S$ as a "treatment" and targeting the average treatment effect on the treated (ATT). This preserves all RCT samples and matches 335 NCDB samples. Distributional balance for the baseline covariates and the estimated sampling score $\hat{\mathbb{P}}(S = 1|X)$ improves significantly after matching, with a visual comparison in Figure 15. However, certain covariates, such as tumor size, remain imbalanced, which could not be addressed by matching without resorting to methods that would undesirably discard RCT samples. This motivates the use of the doubly robust estimator in Sections 2.1 and 3. Notably, while a doubly robust estimator alone can address covariate imbalance, matching as a pre-processing step reduces reliance on correct model specification (Ho et al., 2007). A summary table of the pre-processed data is in Table 2.

**Selection performance.** Figure 16 shows that, given the observed confounder $X$, CSB tends to select ECs whose outcomes are more similar to randomized controls, reducing hidden bias that cannot be addressed by balancing $X$ alone.

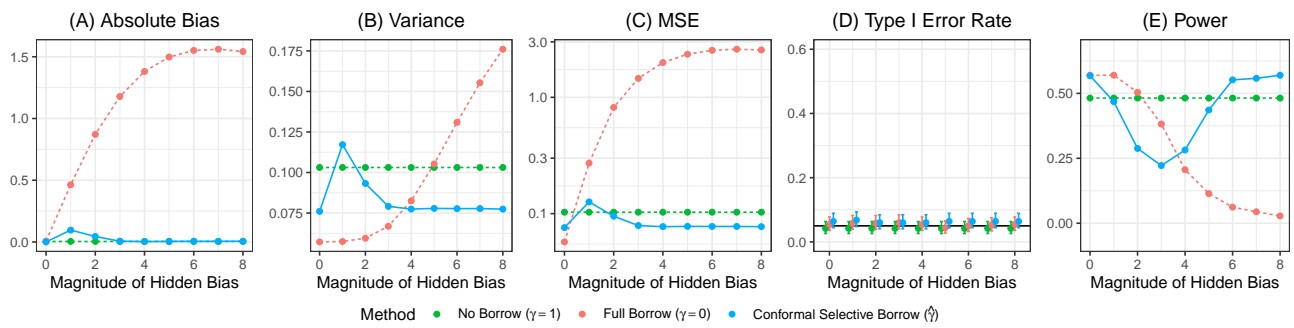

Figure 8. Simulation results when $n_{\mathcal{E}} = 300$. ALSB's exact $p$-value is unavailable due to computation.

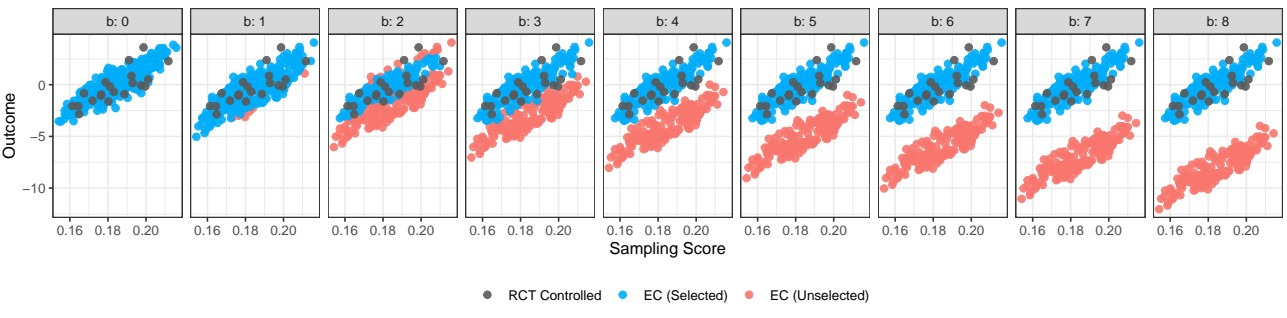

Figure 9. Selection performance of CSB ($\hat{\gamma}$) when $n_{\mathcal{E}} = 300$.

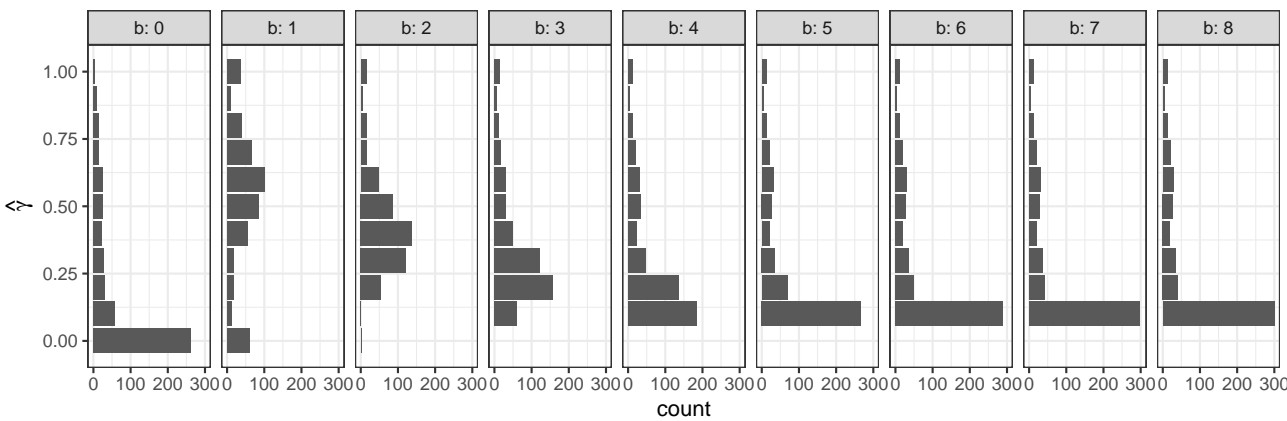

Figure 10. $\hat{\gamma}$ versus $b$ when $n_{\mathcal{E}} = 300$.

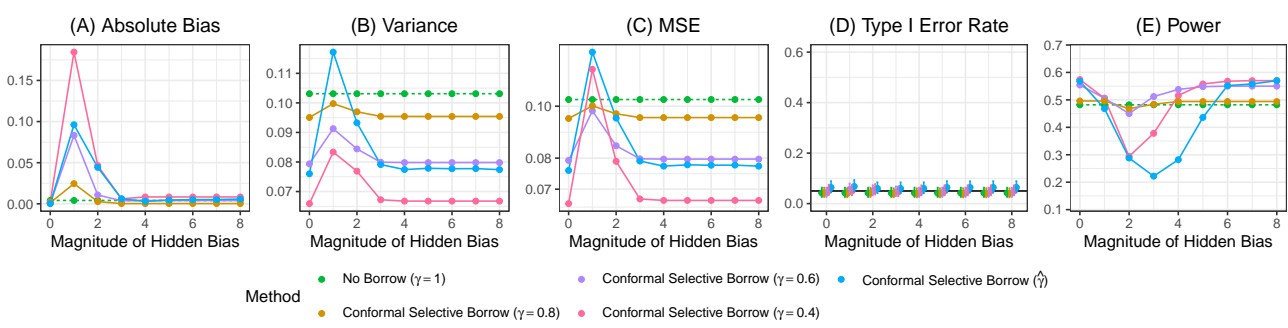

Figure 11. Simulation results for various selection threshold $\gamma$'s when $n_{\mathcal{E}} = 300$.

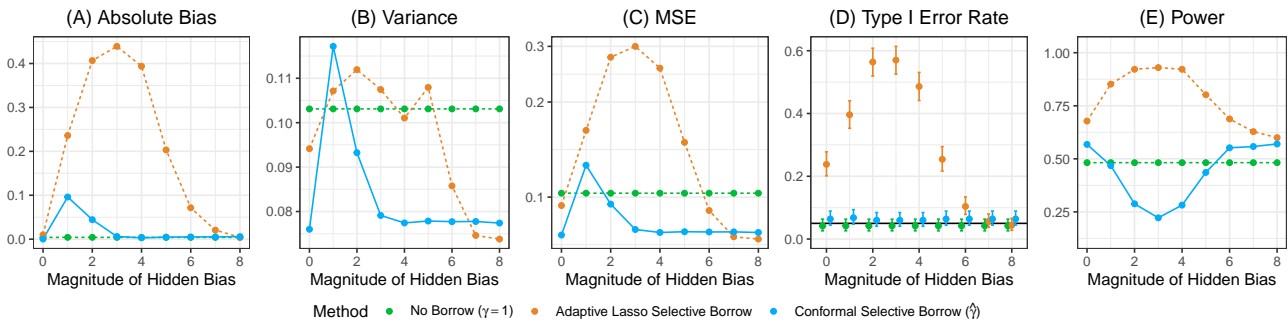

*Figure 12.* Comparison of CSB + FRT and ALSB + asymptotic inference when $n_{\mathcal{E}} = 300$.

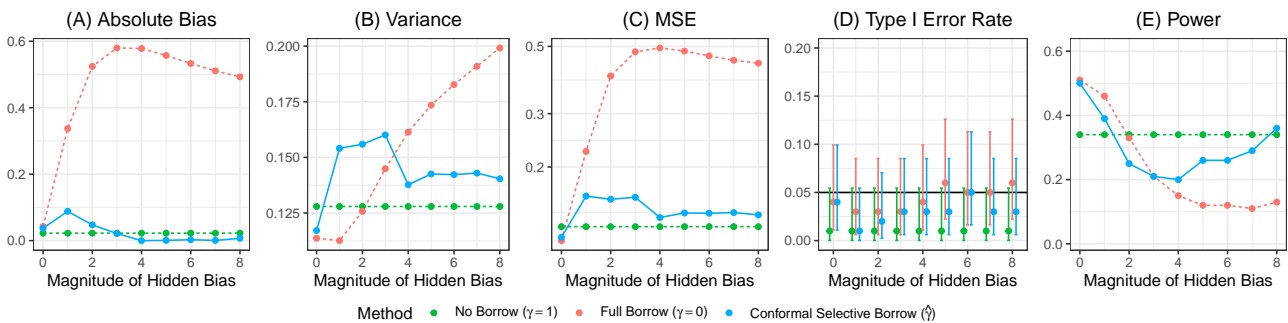

*Figure 13.* Simulation results across different hidden bias magnitudes $b$ for dependent covariates with $p = 5$.

*Table 2.* Summary statistics of the pre-processed data.

|  | C9633 Treated ($n_1 = 167$) | C9633 Controlled ($n_0 = 168$) | NCDB Controlled ($n_{\mathcal{E}} = 335$) |
|---|---|---|---|
| **Sex** |  |  |  |
| Male | 109 (65.3%) | 106 (63.1%) | 219 (65.4%) |
| Female | 58 (34.7%) | 62 (36.9%) | 116 (34.6%) |
| **Age (years)** |  |  |  |
| Mean (SD) | 60.4 (10.2) | 61.2 (9.28) | 60.8 (9.69) |
| Median [Min, Max] | 61.0 [34.0, 78.0] | 62.0 [40.0, 81.0] | 61.0 [34.0, 80.0] |
| **Race** |  |  |  |
| White | 151 (90.4%) | 148 (88.1%) | 300 (89.6%) |
| Non-white | 16 (9.6%) | 20 (11.9%) | 35 (10.4%) |
| **Histology** |  |  |  |
| Squamous | 66 (39.5%) | 65 (38.7%) | 131 (39.1%) |
| Other | 101 (60.5%) | 103 (61.3%) | 204 (60.9%) |
| **Tumor Size (cm)** |  |  |  |
| Mean (SD) | 4.60 (2.04) | 4.56 (2.05) | 4.77 (1.42) |
| Median [Min, Max] | 4.00 [1.00, 12.0] | 4.00 [1.00, 12.0] | 4.50 [3.10, 12.0] |
| **Outcome: 3-year RMST\*** |  |  |  |
| Mean (SD) | 2.77 (0.596) | 2.64 (0.720) | 2.43 (0.947) |
| Median [Min, Max] | 3.00 [0.383, 3.00] | 3.00 [0.181, 3.00] | 3.00 [0.0242, 3.00] |

\*Pseudo-observations transformed from censored survival time.

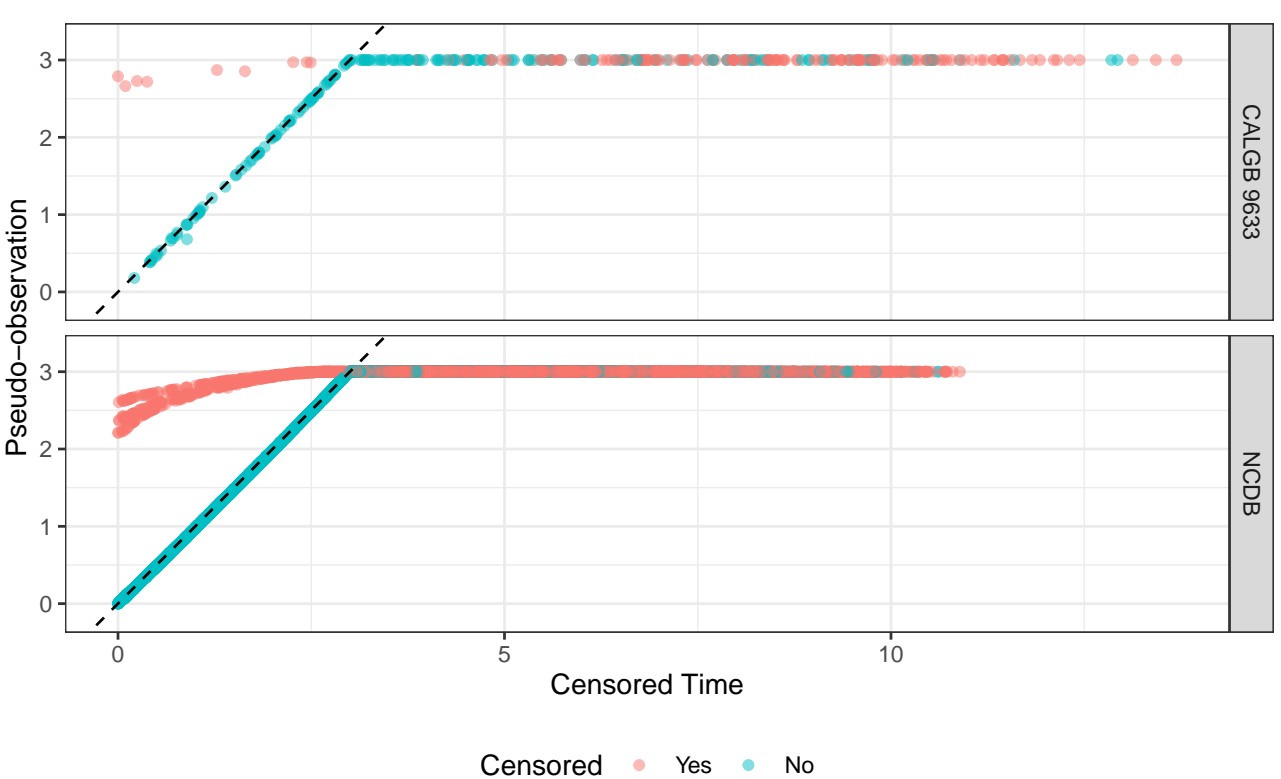

*Figure 14.* Pseudo-observation vs. Censored Time for CALGB 9633 and NCDB datasets.

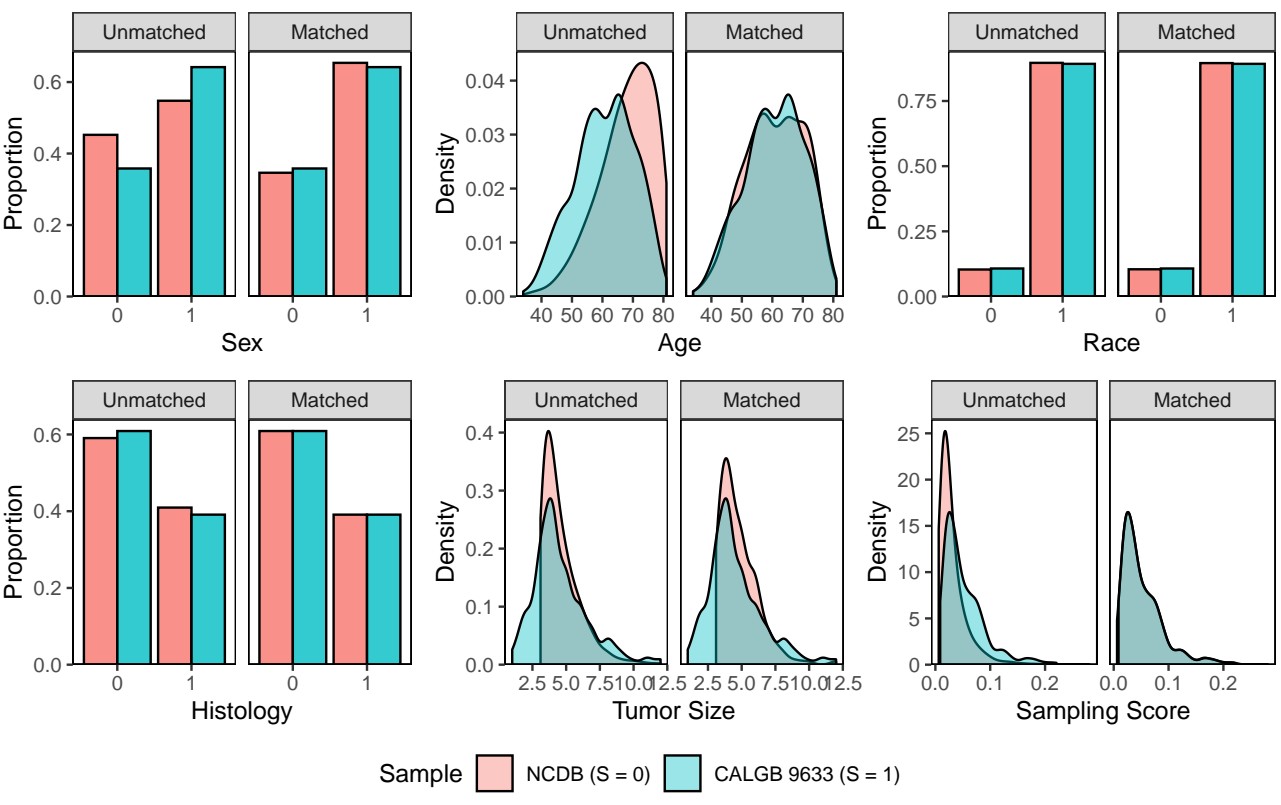

*Figure 15.* Distributional balance (unmatched and matched) between CALGB 9633 ($S = 1$) and NCDB ($S = 0$) for baseline covariates and the estimated sampling score $\hat{\mathbb{P}}(S = 1|X)$.

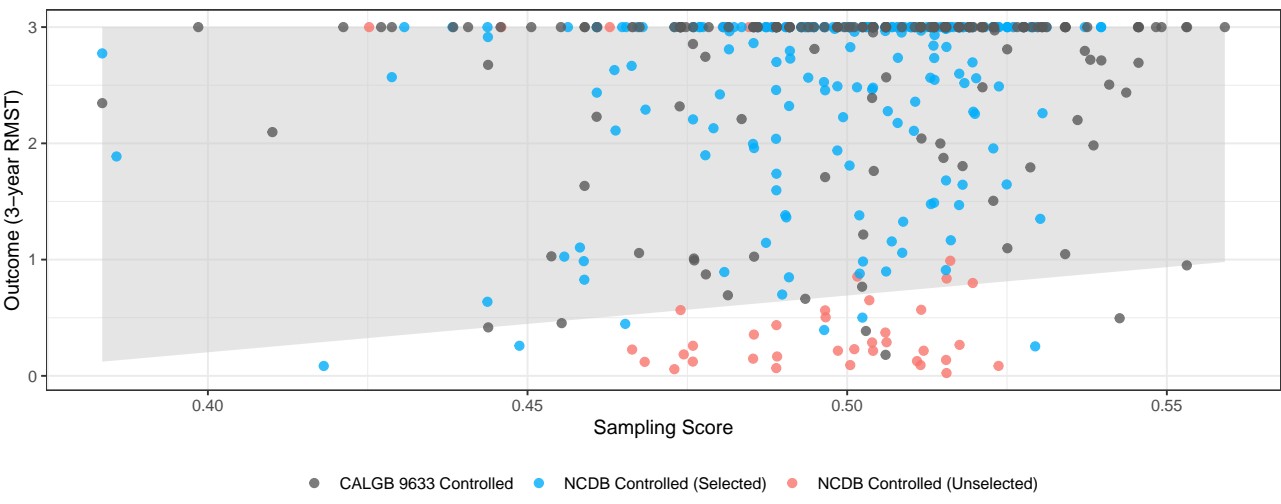

*Figure 16.* 3-year RMST (Outcome) vs. Sampling Score estimated by 5 covariates. The shaded area is constructed using quantile regression on the CALGB 9633 controlled data.

