# OpenReview forum: "Enhancing Statistical Validity and Power in Hybrid Controlled Trials: A Randomization Inference Approach with Conformal Selective Borrowing"
_ICML.cc/2025/Conference — ICML 2025 poster_

### Official Review · Reviewer_uA9f · 2025-03-08

**Overall Recommendation:** 3

**Summary:**

This paper proposes to use Fisher Randomization Test (FRT) in RCTs when leveraging external controls (EC). Since FRT only uses the randomization distribution of a test statistic under the sharp null, it always provides valid type-I error control regardless of how the potentially biased ECs are incorporated. In this way, ECs are used as a tool for constructing more powerful test statistics. In doing so, the authors recognize the issue of bias in ECs, and propose to improve the test statistic by selectively borrowing ECs by thresholding their conformal p-values. To choose the threshold, a cross validation paradigm is also proposed. The proposed methods are demonstrated via simulations and real data illustrations.

**Claims And Evidence:**

Yes.

**Essential References Not Discussed:**

I didn't find important ones to my knowledge.

**Experimental Designs Or Analyses:**

I checked the simulation setups and they seem sound.

**Methods And Evaluation Criteria:**

Yes.

**Other Comments Or Suggestions:**

N/A

**Other Strengths And Weaknesses:**

Strengths:
- The results are rich.
- Propose the importance of FRT in RCT analysis and suggest a new way of using ECs.

Weakness:
In general, this paper contains a lot of results but the motivation and results can be organized in a better way. I list some confusions when reading the paper:
- The comparison with existing methods which motivates the method is not fully reasonable. In particular, the hybrid doubly robust estimators are aimed for providing more accurate estimators, but the authors propose to switch gears to a randomization test for the sharp null, which makes it not fully comparable with the existing method. This makes the arguments that motivate the current proposal a bit weak.
- The definition of conformal p-values are not new in this paper, and it takes too much space in the paper.
- I'm not sure what is the connection between the MSE of $\hat\tau$ and the power of the resulting RCT. In reading the paper, I sometimes felt like the authors wanted to improve MSE of the test statistics but sometimes thought they want to improve the power of RCT. I would suggest the authors clean up the storyline.

**Questions For Authors:**

- Is there a connection between MSE of $\hat\tau$ and power of RCT?
- Is it possible to get back to the estimation problem based on this approach?

**Relation To Broader Scientific Literature:**

- Propose the use of FRT in borrowing information from ECs in RCT analysis.
- Demonstrate the use of selective borrowing in developing more powerful test statistics.
- Demonstrate the use of conformal inference in selective borrowing.

**Theoretical Claims:**

I briefly checked the proofs which seem correct to me.

---

> ### Author Rebuttal · Authors · 2025-03-31
>
> We sincerely appreciate your insightful and constructive comments. Below, we have provided our detailed, point-by-point responses.
>
> **1. Connection between MSE of $\hat{\tau}_\gamma$ and power of FRT**
>
> (i) Variance and power: Theorem 2.4 (power analysis for consistent test statistics) shows that a lower variance of a consistent test statistic leads to a higher power of FRT.
>
> (ii) Bias and power: Simulation-based power analysis demonstrates that borrowing biased ECs can severely reduce FRT power.
>
> (iii) MSE and power: Taking both variance and bias into account, we observe that a lower MSE of the test statistic is associated with higher FRT power. While it remains theoretically challenging to link MSE and power when the test statistic is irregular, our experiments show that MSE-guided $\gamma$ selection offers a unified solution that performs well for both estimation accuracy and power.
>
> We also revised our storyline as follows:
>
> Our primary goal is to improve the power of the Fisher Randomization Test (FRT) for testing the null hypothesis of no treatment effect by borrowing external controls (ECs), compared to relying solely on RCT data (No Borrowing). This is guided by three key insights: (i) Theorem 2.4 shows that the power of FRT can be increased by reducing the variance of a consistent test statistic, which can be achieved by borrowing unbiased ECs to augment the small RCT control sample; (ii) borrowing biased ECs leads to inconsistency and significantly reduces the power of FRT; (iii) these insights motivate our use of Conformal Selective Borrowing to enhance power by borrowing unbiased ECs and discarding biased ones from a larger EC pool.
>
> To this end, we introduce an intermediate objective: testing the exchangeability of each EC and selectively borrowing those deemed unbiased. We recognize that the power of this intermediate testing step is limited by the size of the RCT control group. To address this, we propose tuning the selection threshold $\gamma$ (the significance level of the exchangeability test) to directly target our primary objective. Although tuning based on empirical FRT power would be ideal, it is impractical because (i) it requires specifying an alternative hypothesis and (ii) it is computationally expensive to compute power across a grid of $\gamma$ values.
>
> As a practical alternative, we use the MSE of the Conformal Selective Borrowing estimator as a proxy to guide $\gamma$ selection. This MSE-guided approach offers several benefits: (i) experiments show it improves FRT power over RCT-only analysis, even though the theoretical connection between MSE and FRT power is challenging due to the irregular nature of the test statistic; (ii) it yields strong selection performance and supports our intermediate objective (see Fig. 2 and Fig. 14); (iii) with MSE-guided $\gamma$, CSB serves as both a powerful test statistic and a accurate ATE estimator; (iv) empirical MSE can be approximated leveraging the No Borrowing estimator, and we provide a non-asymptotic excess risk bound for the adaptive procedure.
>
> **2. Conformal selective borrowing (CSB) as a powerful test statistic and a reliable, efficient estimator**
>
> Although our primary motivation is to improve the power of FRT, the final method, CSB with MSE-guided $\gamma$ selection, applies to both estimation and testing:
>
> (i) Theorems 3.7 and 3.8 provide non-asymptotic excess risk bounds for the proposed estimator.
>
> (ii) In simulations, CSB shows better estimation performance than adaptive lasso selective borrowing (ALSB) in terms of MSE under small and moderate sample sizes. This is demonstrated in Figures 6(C) and 11(C), where both methods are compared purely on estimation accuracy without involving inference procedures.
>
> (iii) In real-world experiments, CSB improves both robustness and efficiency over No Borrowing (NB) and Full Borrowing (FB). Specifically, CSB reduces the standard error by 20% compared to NB and mitigates the bias of FB. The ATE estimate under CSB (0.138) is close to NB (0.142), while FB overestimates it (0.241).
>
> These results suggest that CSB is not only effective for testing under sharp nulls but also serves as a reliable and efficient estimator of treatment effects.
>
> **3. Presentation**
>
> Following the reviewer’s suggestion, we moved the full conformal $p$-value (which is computationally infeasible) and jackknife+ (a special case of CV+) to the appendix. We kept the split conformal $p$-value and CV+ $p$-value in the main text, as both are used in our experiments.

---

### Official Review · Reviewer_U7hX · 2025-03-09

**Overall Recommendation:** 3

**Summary:**

This paper proposes a method for combining (potentially biased) external controls (ECs) with data from a randomized control trial, in an effort to improve power to detect causal effects, without sacrificing Type 1 error (false positive) control.  For controlling Type 1 errors, the key insight is to use a Fisher Randomization Test (FRT), which allows for computation of exact p-values for Fisher's "sharp null" hypothesis (potential outcomes are constant across treatment/control) with any test statistic, including those using potentially biased ECs.  For improving power, there is no "free lunch" (as noted by the authors and other related work), but an approach is proposed for "selective borrowing" of less biased ECs, which is illustrated empirically in simulated data and a real-data application to a chemotherapy randomized trial.

**Update after rebuttal**: As stated below, I will keep my score.  My impression of the paper is somewhere between a 3 and a 4.

**Claims And Evidence:**

The main claims as I understood them:
* Their approach will have valid level / false positive rate in finite samples for the Fisher sharp null.
* Their power can be better than "no borrowing" (i.e., just using the RCT) if the bias is small
* Their approach is better suited to finite samples than other selective borrowing approaches like Gao et al. 2023.

The claims are supported by clear and convincing arguments, and backed up by synthetic experiments where the bias can be controlled.  I did find one claim a little difficult to understand (regarding post-selection inference), which I have mentioned in "Questions for the Authors" below, but I think this is a minor point.

**Essential References Not Discussed:**

Overall, I found that the related work (in the Appendix) was quite comprehensive, but would have appreciated having more of that content in the main paper, if space permits.

**Experimental Designs Or Analyses:**

The experimental design for the synthetic data seems sound to me, and is broadly similar to how I would have set up a synthetic data experiment for this method, probing failure modes as a function of the bias. The real-data experiment is more an "illustrative application / case study" rather than an experiment that tests a particular hypothesis, but that's a fairly standard approach for these types of papers in my experience.

**Methods And Evaluation Criteria:**

Yes, for the most part (one important missing baseline, excluded due to computational concerns). See my comment on "relation to the broader scientific literature"

**Other Comments Or Suggestions:**

Some minor grammar / presentation points:
* On Line 056, right-hand column: "Let the binary treatment denote by $A$" -> "Let $A$ denote the binary treatment"
* Figure 1 is extremely small.  Note that you can use figure* in two-column formats to get a figure that crosses both columns, which would be better for this figure.

**Other Strengths And Weaknesses:**

Beyond the lack of direct apples-to-apples comparison with ALSB, I have two (relatively minor) concerns regarding clarity / significance.

1. I'm not sure how to square the null hypothesis tested under the FRT (the "sharp null") with the more conventional null hypothesis that e.g., the average treatment effect is equal to zero, which doesn't require that $Y_1 = Y_0$, but rather just that $E[Y_1] = E[Y_0]$.  Hence, while the FRT controls the false discovery rate under the sharp null, it's not clear that it controls the false discovery rate under more conventional null hypotheses.  It may be worth clarifying this point in the paper, or otherwise commenting on what is "lost" by relying on FRT.
2. It's not clear to me how novel / original some of the theoretical results are, and it would be worth clarifying which theoretical results are considered most novel by the authors.  Theorem 2.3, for instance, seems like an obvious consequence of FRT, and I'm surprised that it's given as a Theorem in this paper, as opposed to being cited from elsewhere (though given my lack of familiarity with the FRT literature, I don't have a citation to provide).

**Questions For Authors:**

I have listed these in priority order, with the first two questions being particularly important to clarify.

1. I am less familiar with randomization inference (a la the FRT).  Does this procedure also provably control the false positive rate under the more conventional general null hypothesis that $E[Y_1 - Y_0] = 0$?  I would assume not, since it takes $Y_1, Y_0$ as fixed, as opposed to random?
2. How does CSB compare to ALSB under an apples-to-apples comparison?  For context: The comparison in Figure 6 is not "apples to apples", instead it shows FRT + CSB versus ALSB with asymptotic inference.  It would be nice to see either (a) FRT+CSB versus FRT+ALSB, or (b) CSB and ALSB considered head-to-head where both use asymptotic inference.
3. Which of the theoretical results are the most novel, in the view of the authors?  To me, it seems like most of the theory is relatively straightforward, and the main contributions are a bit more conceptual (e.g., the realization that one can use the FRT here), but I'm open to arguments that there are more novel theoretical contributions here.
4. What is the concern with "post-selection inference" here?  I don't quite follow the claim (see 090-092 LHS, "We account for selection uncertainty in FRT and offer valid post-selection inference").  The only justification I see is on lines 222-225 (LHS), "By using $T(A) = |\hat{\tau}_{\gamma}|$ as the test statistic for FRT and allowing $\mathcal{E}(\gamma)$ to vary with resampling A in FRT could account for selection uncertainty and provide valid post-selection inference". This is mainly just a conceptual argument for why there is no post-selection inference concern, right?  I.e., under the null, the entire procedure is run to get a sampling distribution of the test statistic, which can depend in arbitrary ways on the data?

**Relation To Broader Scientific Literature:**

This paper adds yet another method to a growing literature (well-documented in Section A) on combining observational and experimental data in an adaptive fashion, without making assumptions on the validity of the observational data.

The core insight of this paper, in my view, is that Fisher Randomization Tests can be used with any test statistic, and therefore they are a good candidate for "improvement" with external controls, since the false positive rate is controlled exactly regardless of how good the test statistic is.

Similar to other work in this area, this paper identifies, at least experimentally, a no "free lunch" phenomenon, , i.e., there are moderate biases which can lead to loss of performance. However, beyond some intuition-building theory (e.g., Theorem 2.4), the authors do not formally analyze this relationship between bias and power.  That said, I am sympathetic to the difficulty in doing so.

It is harder to judge the significance of the conformal selective borrowing approach.  While it is novel, it's not clear (on it's own, without the FRT component), how this approach compares to other selective borrowing approaches, especially given the lack of experimental comparison (on an apples-to-apples basis) with the Adaptive Lasso Selective Borrowing (ALSB) approach.

**Theoretical Claims:**

I did not see any issues in the proofs. I checked the proof of Theorem 2.3, which seemed fairly immediate from the definition of the FRT, and Theorem 2.4, which similarly seems to follow fairly directly from auxiliary lemmas from related work (which I did not check).  While I did not review the proofs of Propositions 3.1-3.6 in depth, they seem to follow from standard arguments, though I'm a little less familiar with the conformal prediction literature.  Theorems 3.7 and 3.8 follow from standard results (and some algebra, which I did not check in detail) in non-asymptotic statistics.

---

> ### Author Rebuttal · Authors · 2025-03-31
>
> We sincerely appreciate your insightful and constructive comments. Below, we have provided our detailed, point-by-point responses.
>
> **1. Clarification on sharp null hypothesis**
>
> The sharp null hypothesis $Y_i(1) = Y_i(0)$ for all $i \in$ RCT states that there is no individual treatment effect for any unit. Another common form of the sharp null is conditional independence, $Y_i \perp A_i \mid X_i$, meaning the observed outcome is independent of treatment assignment given covariates. In contrast, the hypothesis that the average treatment effect (ATE) is zero is often called the "weak null."
>
> In a finite-sample exact sense, FRT guarantees Type I error control under the sharp null but cannot guarantee it under the weak null. However, recent work has shown that FRT can also asymptotically control the Type I error under the weak null using studentized or pre-pivoted test statistics (Wu & Ding, 2021; Cohen & Fogarty, 2022). The RCT sample size is typically small in our context, such as trials for rare diseases. Asymptotic approximations may be unreliable, so we rely on the sharp null for a finite-sample exact test.
>
> While this reliance on the sharp null rather than the weak null may be viewed as a limitation of FRT, to our knowledge, there is currently no testing procedure that controls Type I error in a finite-sample exact sense under the weak null without additional distributional assumptions. Section 6 also discusses possible future directions beyond the sharp null.
>
> **2. Significance of the Conformal Selective Borrowing (CSB) and comparison to Adaptive Lasso Selective Borrowing (ALSB)**
>
> Compared to existing selective borrowing approaches such as ALSB, the significance of CSB can be summarized as follows:
>
> (i) Model-free flexibility: CSB is a model-free approach that allows flexible choice of conformal scores depending on data characteristics. For example, in our real-world application where the outcome exhibits heavy tails and heteroscedasticity (see Figure 14), we use conformalized quantile regression (Romano et al., 2019) for selection. Distance-based scores such as nearest-neighbor conformity scores (Shafer & Vovk, 2008) can be used for binary outcomes. This flexibility is difficult to achieve with model-based methods like ALSB.
>
> (ii) Estimation: CSB performs better than ALSB in terms of MSE under small and moderate sample sizes. This is demonstrated in Figures 6(C) and 11(C), where both methods are compared purely on estimation accuracy without involving inference procedures.
>
> (iii) Computation: CSB is compatible with the FRT, while ALSB is not readily applicable with FRT due to its computational complexity. This highlights an advantage of CSB when exact finite-sample inference is desired.
>
> (iv) Apples-to-apples comparison under asymptotic inference (Asym): We conducted a comparison between CSB+Asym and ALSB+Asym (see [here](https://anonymous.4open.science/r/doc-E6B4/sim_csb_asym.pdf)). CSB+Asym generally achieves better Type I error control than ALSB+Asym and performs comparably when $b=1$.
>
> **3. Novelty and implications of theoretical results**
>
> Theorem 2.3 builds on classical FRT results (e.g., Lehmann and Romano, 2005, Theorem 15.2.1), but to our knowledge, this is the first formal result establishing the validity of FRT in the context of hybrid controlled trials. It confirms that FRT can be applied in this setting and clarifies how it should be applied to ensure validity. Specifically, Theorem 2.3 provides the following practical guidance and important caveats:
>
> (i) Type I error control is guaranteed only when we permute assignments according to the actual experimental design, that is, permuting treatment assignments only within the RCT while keeping the EC assignments fixed. An important caveat is that permuting across all treated and control units, including ECs, would invalidate the theorem.
>
> (ii) To maintain validity, the test statistic should vary with the permuted treatment vector $A$, meaning the selected set of ECs should also be updated under each permutation to account for selection uncertainty. Fixing the EC selection across permutations would also invalidate the theorem.
>
> Theorems 3.7 and 3.8 provide novel non-asymptotic MSE bounds that guide the design of the adaptive selection threshold in finite samples. Please also see our response to Reviewer f66S, Point 1.
>
> **4. Post-selection inference**
>
> The reviewer's understanding is correct. The selection uncertainty is fully incorporated into the reference distribution by allowing the selected set to vary with the resampled treatment vector $A$ in the FRT. Therefore, there is no post-selection inference concern when following this principle. However, as noted in the previous point, a key caveat is that the selected EC set should not be fixed during permutation, as this would ignore selection uncertainty and invalidate the test.
>
> **5. Presentation**
>
> We moved the related work to the main text, revised Line 056, and enlarged Figure 1.

---

> > ### Comment · Reviewer_U7hX · 2025-04-07
> >
> > Thank you for the response - those clarifications are very helpful.  I am inclined to keep my score (I'm somewhere between a 3 and a 4 after the response).  Reading the review of GG5D, I would also suggest being more explicit upfront that there is no free lunch here, that you have proposed a valid approach that "might" improve power, if all goes well.

---

> > > ### Author Response · Authors · 2025-04-07
> > >
> > > Thank you very much for your thoughtful follow-up and valuable suggestions. We're pleased that your score has risen to 3.5, even though the official score may not reflect this according to the new scoring scale.
> > >
> > > We fully agree that it is important to be upfront about the limitations. Our proposed method controls the Type I error for testing the sharp null and might improve the power of RCT-only analysis when the bias of ECs is either absent or detectable. When the bias is difficult to detect, our method may incur some power loss, though it still maintains valid Type I error control.
> > >
> > > We also appreciate the importance of positioning our method within the context of the existing literature. The no-free-lunch limitation is recognized in existing review papers (Oberst et al., 2022; Lin et al., 2024), which point out that no method can uniformly and significantly outperform RCT-only analysis across varying levels of hidden bias, although different approaches optimize the risk-reward trade-off from different perspectives. The most challenging scenarios are those where bias is present but complex to correct or difficult to detect. Our main difference from existing literature is twofold: (i) we prioritize exact Type I error control in small samples first, then seek to improve power; (ii) we optimize the risk-reward trade-off between no borrowing and full borrowing from the perspective of conformal selective borrowing, motivated by our real data, where some ECs are unbiased while others are not.
> > >
> > > We will incorporate this important discussion more explicitly in the introduction section to better align with your and Reviewer GG5D’s helpful feedback.

---

### Official Review · Reviewer_f66S · 2025-03-14

**Overall Recommendation:** 3

**Summary:**

This paper proposes a randomization inference framework that can combine the data from randomized controlled trials with external controls. The proposed method controls the Type-I error in finite samples by leveraging conformal inference to select appropriate samples from external controls. In particular, the selection threshold provides an interpolation between no-borrowing and full-borrowing approaches, which can be tuned by minimizing the mean squared error. Some simulation results and real-world applications show the applicability of the proposed method.

**Claims And Evidence:**

Basically, the claims in the submission are clear and supported by proofs or simulation studies. However, I feel that the bounds in Theorems 3.7 and 3.8 are loose and not easy to comprehend.

In addition, I don't quite understand why the authors can relate $\hat{\tau}_{\hat{\gamma}}$ to super efficiency on the second column of Line 295. Can the authors provide some theoretical justification to this claim?

**Essential References Not Discussed:**

Not that I am aware of.

**Ethical Review Concerns:**

The paper exceeds the 8-page limits and potentially discloses the authors' identities.

**Ethical Review Flag:**

Flag this paper for an ethics review.

**Ethics Expertise Needed:**

["Other expertise"]

**Experimental Designs Or Analyses:**

I checked the experimental results and already pointed out my concerns for the simulation setups above.

Moreover, for Figures 5 and 10, the authors should also report the value of $\hat{\gamma}$. The reason is that the proposed method CBS under the "optimal" value $\hat{\gamma}$ performs not as good as the cases under other values of $\gamma$. This is a severe issue that may limit the impact of this paper.

**Methods And Evaluation Criteria:**

The proposed methods make sense to me. However, for the simulation setups, the dimension of covariates is $p=2$, which is too small and won't lead to meaningful conclusions. Can the authors try some larger values for $p$? Also, the current covariates $X$ have independent coordinates. Can we introduce any dependence within the coordinates of $X$?

**Other Comments Or Suggestions:**

1. Second column of Lines 178 to 182: The sentence "Based on..." seems to repeat what have been discussed in the last paragraph.

2. Second column of Line 225: There should be a division in the definition of $p_j^{split}$. The same issues happen to $p_j^{jackknife+}$ and $p_j^{cv+}$.

3. The plot in Figure 2(D) needs to be zoomed in.

**Other Strengths And Weaknesses:**

As I mentioned above for the simulation results, the proposed method under the optimal choices of selection parameter does not seem to perform as good as the case under other arbitrary choices.

**Questions For Authors:**

See above.

**Relation To Broader Scientific Literature:**

This paper combines the approaches in conformal inference literature with other randomization tests to propose a new framework for combining the data from randomized controlled trials with external controls. The estimator relies on the one proposed by Li et al. (2023), but the authors also relax the mean exchangability condition imposed by Li et al. (2023).


Li, X., Miao, W., Lu, F., and Zhou, X.-H. Improving efficiency of inference in clinical trials with external control data. Biometrics, 79(1):394–403, 2023.

**Theoretical Claims:**

I basically checked all the proofs, and they looks correct. A minor question is how $\Phi$ is related to variances of $\epsilon_{\gamma}$. Can the authors provide some short discussion on it?

---

> ### Author Rebuttal · Authors · 2025-03-31
>
> We sincerely appreciate your insightful and constructive comments. Below, we have provided our detailed, point-by-point responses.
>
> **1. Bound in Theorems 3.7 and 3.8**
>
> We explain the key terms in Theorems 3.7 and 3.8 and their practical implications as follows:
>
> (i) The term $c\Delta|\delta_1|$ involves $\delta_1$, the bias of the consistent No Borrowing estimator, which is of order $1/n$, and $\Delta$, the maximum bias across all $\gamma \in \Gamma$. This motivates us to apply pre-propensity-score matching on all ECs to prevent potentially large bias during the data preparation phase.
>
> (ii) The term $c\Delta\Phi\sqrt{\log (|\Gamma|/\iota)}$ involves $\Phi$, the largest standard deviation proxy $\phi$ across $\gamma \in \Gamma$, which is of order $O(1/\sqrt{n})$. The grid size $|\Gamma|$ is fixed (e.g., 10 in our experiments), making this term well-controlled.
>
> (iii) The term $\\max (c\Phi^2\sqrt{\log (|\Gamma|/\iota)}, c\Phi^2\log (|\Gamma|/\iota))$ arises from sub-exponential tail bounds and scales similarly to (ii).
>
> (iv) The terms $\\max\_{\gamma\in\Gamma} |\hat{V}(\hat{\tau}\_{\gamma} - \hat{\tau}\_1) -\kappa_{\gamma}^2|$ and $\\max\_{\gamma\in\Gamma} |\hat{V}(\hat{\tau}\_{\gamma}) - \sigma^2_{\gamma}|$ corresponds to the estimation errors for $\kappa^2_{\gamma}$ and $\sigma^2_{\gamma}$. This motivates the use of a sufficiently large number of bootstrap replicates to ensure accurate variance estimation.
>
> Overall, Theorems 3.7 and 3.8 provide concrete guidance for implementing $\gamma$ selection in practice and justify the design of our MSE-guided adaptive procedure, even if the bounds themselves are conservative due to their non-asymptotic and worst-case nature.
>
> **2. Clarifying the connection to super-efficiency-type behavior**
>
> The reviewer questions the super efficiency of $\hat{\tau}_{\hat{\gamma}}$. We acknowledge that this was a misstatement; our intent was not to claim that it is super-efficient, but rather that it exhibits behavior similar to the Hodges estimator. We have revised the sentence as follows:
>
> "This phenomenon highlights that $\hat{\tau}_{\hat{\gamma}}$ behaves similarly to the Hodges estimator (Le Cam, 1953) and to integrated estimators in data fusion (Yang et al., 2023; Oberst et al., 2022): improving upon the baseline estimator (here, the No Borrow estimator) in certain regions of the parameter space (where there is no bias in ECs) inevitably leads to worse performance in other regions (where the bias in ECs is difficult to detect)."
>
> **3. Additional simulations**
>
> We additionally consider $p = 5$ and $X \sim N(0, \Sigma)$, where $\Sigma$ is a Toeplitz matrix with $(i,j)$-th entry $\Sigma_{ij} = \rho^{|i-j|}$ and $\rho = 0.3$, to introduce dependence among the coordinates of $X$. We did not consider larger $p$ since the sample size is small, with only 25 RCT controls. The simulation results (see [here](https://anonymous.4open.science/r/doc-E6B4/sim_supp.pdf)) show similar patterns and demonstrate the robustness of our method.
>
> **4. $\Phi$ and $\epsilon_\gamma$**
>
> $\Phi$ is the largest standard deviation proxy $\phi$ over $\gamma \in \Gamma$. If $\epsilon_\gamma$ is Gaussian, then $\phi$ equals its standard deviation. We previously mislabeled $\phi$ as a variance proxy and have corrected this.
>
> **5.  Value and performance of $\hat{\gamma}$**
>
> Since $\hat{\gamma}$ is adaptive and varies across simulation replicates, we separately report its values in Figure 4 and analyze its behavior in Section C.2.
>
> We do not expect $\hat{\gamma}$ to outperform all fixed $\gamma$ values uniformly. Prior work has shown that no method can uniformly outperform No Borrowing (corresponding to $\gamma = 1$) without additional assumptions (Oberst et al., 2022; Lin et al., 2024). Instead, our proposed $\hat{\gamma}$ is designed to improve power when the bias of ECs is either absent or detectable. When the bias is difficult to detect, it is reasonable that our method may incur some inevitable power loss, though within an acceptable range. Importantly, Conformal Selective Borrowing+FRT always controls the Type I error, even in such challenging cases.
>
> **6. Presentation**
>
> We removed the sentence "Based on...", corrected the missing division symbols, and zoomed in Figure 2(D).
>
> **7. Ethical issue**
>
> As per ICML guidelines, the Impact Statement is not counted toward the 8-page limit. We initially understood this to apply to the Software and Data section and had no intention of violating the page limit.
>
> We did not disclose any identities or URLs in the submission. We mentioned the availability of the R package solely to highlight the applicability and reproducibility of our work. We sincerely apologize if our identity may have been inadvertently revealed in the package documentation, which is external to the submission; this was entirely unintentional and not meant to compromise anonymity. To avoid any potential concern, we have removed the Software and Data section from the manuscript.

---

### Official Review · Reviewer_GG5D · 2025-03-17

**Overall Recommendation:** 2

**Summary:**

Authors study integration of external (historical) controls into randomized controlled trials in a principled way using conformal p-values, to guarantee type-1 error rates with finite samples under potential violation of the exchangeability assumption between trial and external controls.

### update after rebuttal

I thank the authors for their detailed response. In particular, the motivation/logic they outline in the beginning through 3 items: (i), (ii), (iii) are helpful and a more detailed and polished version should definitely be included in the updated manuscript. While those items helped me understand the contributions of the manuscript a little better, I still have concerns regarding sample splitting in the RCT to test ECs against: this  decreases power (reduced sample size in the RCT, as the samples we use for testing ECs should not be used again in downstream analyses). While there is some empirical evidence suggesting improved power overall, the fundamental tradeoff here should be more clearly analyzed theoretically. Therefore, I maintain my score.

**Claims And Evidence:**

The thing that bothers me the most is that it is not clear what authors claim to do and what they end up doing. I can understand the idea of using conformal inference to "select" external controls based on their conformity score, which is straightfoward. However the following question is not answered sufficiently in my opinion:

- You use RCT controls to select in the first place. In that case, why would you not still be limited statistically by the number of controls in the RCT?

The question above is not answered/discussed verbally or as a theorem. By the latter, I mean the following. Authors claim that their new approach controls the type-1 error and improves power. I do not see a result where they "prove" they improve power. And they should be more clear that this "power" only relates to identifying whether or not the treatment has any effect, and not necessarily what is the size of the effect.

A criticism to this conformal approach would be that each control is tested for pooling individually, and then the high probability guarantees are obtained straightforwardly by union bound. As authors briefly mention, this will suffer from low power and may introduce bias. This to me is the point where the paper is making its main contribution via an improved selection mechanism (correct me if I am wrong). However, my main issue here is the following:

- You motivate minimizing MSE of the estimator as a proxy to guide better selection and leverage consistency of the estimators in doing that. The consistency is prone to the same issues you cite to be immune to: asymptotical validity, etc.
- The results you have here again do not connect to an improved power for the FRT, but rather seem to be self-contained and remain as a proxy/heuristic to improve the power.

Please correct me if I am wrong. My main challenges with this paper is that I do not understand what is the one most important result its trying to prove (higher power for for FRT?) and how does the things you do/prove in the paper connect to that? I think the flow of methodology and the motivation of the paper is not very clear as it stands.

**Essential References Not Discussed:**

There is a vast body of work on historical controls, especially in the epidemiology literature. Current manuscript does a poor job covering that.

**Experimental Designs Or Analyses:**

The conformal selective borrowing approach in the real-world experiments seem to be not making a big difference compared to the no borrowing approach. An exception is slightly reduced p-vals and SE's, which might be a function of sample size alone. This falls in line with my earlier intuititon/concern regarding not having improved power as the selection is still limited by the RCT (See Claims And Evidence).

**Methods And Evaluation Criteria:**

Real-world experiment is well described and interesting. It is a suitable experiment to run for this paper.

**Other Comments Or Suggestions:**

See Claims And Evidence

**Other Strengths And Weaknesses:**

See Claims And Evidence

**Questions For Authors:**

See Claims And Evidence

**Relation To Broader Scientific Literature:**

Current manuscript focuses on a specific problem and claims to improve it using ideas from conformal inference. It does not relate to the broader scientific literature in a significant way.

**Theoretical Claims:**

I did not check the proofs carefully. Most results seem to be standard from the conformal inference literature.

---

> ### Author Rebuttal · Authors · 2025-03-31
>
> We sincerely appreciate your insightful and constructive comments. Below, we have provided our detailed, point-by-point responses.
>
> **1. Clarifying the objective and the role of MSE minimization in improving FRT power**
>
> Our primary objective is to improve the power of the Fisher Randomization Test (FRT), which is limited when using only RCT data (No Borrowing). To achieve this, we rely on three key insights:
>
> (i) Variance and power: Theorem 2.4 (power analysis for consistent test statistics) shows that power can be improved by reducing the variance of a consistent test statistic, which can be achieved by borrowing unbiased ECs to augment the limited sample size of the RCT control arm;
>
> (ii) Bias and power: borrowing biased ECs renders the full borrowing estimator inconsistent, severely reducing the power of FRT;
>
> (iii) Bias-variance trade-off and power: these motivate the use of Conformal Selective Borrowing to improve FRT power of RCT-only analysis by borrowing unbiased ECs and discarding biased ones from a large EC pool.
>
> Based on above insights, we introduce our intermediate objective: testing the exchangeability of each EC. We acknowledge that the number of RCT controls statistically limits the power of this exchangeability testing. Therefore, we propose tuning the selection threshold $\gamma$ (i.e., the significance level of the EC exchangeability test) to optimize our primary objective directly. While using empirical FRT power to tune $\gamma$ would be ideal, this approach (i) requires specifying an alternative hypothesis and (ii) is computationally intensive, as computing FRT power across a grid of $\gamma$ values is costly.
>
> This leads to our final choice: using the MSE of the Conformal Selective Borrowing estimator as a proxy to optimize $\gamma$. Our MSE-guided $\gamma$ selection offers several advantages: (i) experiments show it achieves our primary goal of improving FRT power compared to RCT-only analysis, though we acknowledge that the theoretical link between MSE and FRT power is challenging due to the irregularity of the test statistic; (ii) it yields strong selection performance and supports our intermediate objective (see Fig. 2 and Fig. 14); (iii) with MSE-guided adaptive $\gamma$, Conformal Selective Borrowing serves as both a powerful test statistic and an accurate ATE estimator; (iv) empirical MSE can be approximated by leveraging the No Borrowing estimator, and we provide a non-asymptotic excess risk bound for the adaptive procedure.
>
> **2. Finite-sample validity of FRT**
>
> For the validity of FRT (Type I error control), we do not rely on asymptotic arguments such as estimator consistency, as shown in Theorem 2.3. This is a key advantage of FRT over existing integrative methods whose validity depends on asymptotic theory.
> For power improvement, the optimality of the selected $\gamma$ does rely on the consistency of the No Borrowing estimator, as shown in Theorem 3.8. However, our experiments show that Conformal Selective Borrowing with adaptive $\gamma$ improves FRT power even in small-sample settings.
>
> **3. Significance in real-world experiments**
>
> The real-world experiments show that Conformal Selective Borrowing (CSB) improves both robustness and efficiency over No Borrowing (NB) and Full Borrowing (FB): (i) Given that CSB+FRT theoretically controls Type I error in finite samples, which existing integrative methods do not, it yields a significant result (p < 0.05) compared to the borderline NB result (p = 0.055), addressing the underpower issue in the original study. (ii) CSB reduces the standard error by 20% compared to NB. (iii) CSB mitigates the bias of FB; the ATE estimate under CSB (0.138) is close to NB (0.142), while FB overestimates it (0.241). We acknowledge that the extent of improvement depends on the quality of the real data at hand. To further evaluate our method in finite samples, our simulations show that CSB can improve FRT power by up to 45% compared to NB.
>
> **4. Related work on historical controls**
>
> Due to the manuscript's space limitation at the original submission, we included the comprehensive literature review on historical control borrowing in Appendix A. We acknowledge the value of such a review in the main text and will move the related work on historical controls to the Introduction section at the resubmission. Please let us know if we have missed any relevant literature.

---

### Decision · Program_Chairs · 2025-05-01

**Decision:**

Accept (poster)

**Comment:**

This paper introduces a new method to integrate observational external controls with data from a RCT to estimate causal effects while controlling the risk that the external controls introduce bias into the estimates. Most reviewers found the introduction of Fisher randomization test, alongside an adaptively tuned selective borrowing step, to be worthwhile additions to the literature that would be of interest to others working on the problem of integrating RCTs and observational data. The main concern, raised by one reviewer, is that the tuning in the selective borrowing step targets a proxy objective (MSE) instead of the true power. In my view this is a reasonable choice and the concerns here can be addressed by revising the paper to be clearer about what strategy is being chosen and why.